# Genome-wide identification and comprehensive analysis of the phytochrome-interacting factor (*PIF*) gene family in wheat

**Hua Zhuang[1,2], Zhen Guo[1,2], Jian Wang[1,2,3], Tianqing Chen**[1,2,3,4,5] *

**1** Shaanxi Provincial Land Engineering Construction Group Co., Ltd, Xi'an, China, **2** Institute of Land Engineering and Technology, Shaanxi Provincial Land Engineering Construction Group Co., Ltd, Xi'an, China, **3** Key Laboratory of Degraded and Unused Land Consolidation Engineering, Ministry of Natural Resources, Xi'an, China, **4** Shaanxi Engineering Research Center of Land Consolidation, Xi'an, China, **5** Land Engineering Technology Innovation Center, Ministry of Natural Resources, Xi'an, China

* 369668050@qq.com

**Data Availability Statement:** All relevant data are within the paper and its Supporting Information files.

## Abstract

Phytochrome-interacting factors (PIFs) are essential transcription factors for plant growth, development, and stress responses. Although *PIF* genes have been extensively studied in many plant species, they have not been thoroughly investigated in wheat. Here, we identified 18 *PIF* genes in cultivated hexaploid wheat (*Triticum aestivum* L). Phylogenetic analysis, exon-intron structures, and motif compositions revealed the presence of four distinct groups of TaPIFs. Genome-wide collinearity analysis of *PIF* genes revealed the evolutionary history of *PIFs* in wheat, *Oryza sativa*, and *Brachypodium distachyon*. *Cis*-regulatory element analysis suggested that *TaPIF* genes indicated participated in plant development and stress responses. Subcellular localization assays indicated that TaPIF2-1B and TaPIF4-5B were transcriptionally active. Both were found to be localized to the nucleus. Gene expression analyses demonstrated that *TaPIFs* were primarily expressed in the leaves and were induced by various biotic and abiotic stresses and phytohormone treatments. This study provides new insights into PIF-mediated stress responses and lays a strong foundation for future investigation of *PIF* genes in wheat.

## 1. Introduction

Phytochromes (PHYs), as one of photoreceptors, participate in a series of downstream physiological and biochemical processes, e.g., plant growth, stress, and defense responses [1–5]. PhyB protein integrates multiple signaling pathways to control photomorphogenesis. Additionally, it interacts with subgroup 15 of the basic helix-loop-helix (bHLH) transcription factor (TF) family; hence, these transcription factors are referred to as PHOTOCHROMIN INTERACTION FACTORs (PIFs) [6, 7]. The first PIF protein in *Arabidopsis*, AtPIF3, was identified by yeast two-hybrid experiments using the C-terminal region of PhyB [8]. Subsequent studies revealed that the N-terminal region of the PIF protein contains a conserved active PhyB-binding (APB) motif, and the C-terminal region of the PIF protein

**Funding:** This work was supported by the Shaanxi Province "High-level Talents Special Support Plan" Outstanding talents project, Construction of a shared platform for soil quality detection and evaluation in Shaanxi Province (2021PT-053). The funders had no role in study design, data collection and analysis, decision to publish, or preparation of the manuscript.

**Competing interests:** The authors have declared that no competing interests exist.

contains the bHLH domain, which performs the function of binding to other proteins or DNA [9, 10]. As a member of the bHLH family, PIF proteins can also regulate downstream gene expression by binding G-box (CACGTG) and/or E-box (CANNTG) moieties in target gene promoters [6].

Several PIFs have been identified in agronomically important plants. These include eight identified PIFs in tomato and apple [11, 12], respectively, and six identified PIFs in rice (*Oryza sativa*) and *Brachypodium distachyon* [13, 14], eight reported PIFs in *Arabidopsis* [11], potato [15], and maize (*Zea mays*) [16], respectively. Recent studies have shown that PIF proteins are broadly engaged with plant growth, development, and stress responses [17–21]. In terms of growth and development, PIF proteins have been associated with the guideline of seed germination [22, 23], blossoming time [24, 25], shade avoidance [26], and circadian processes [18]. For example, AtPIL5/PIF1 regulate the expression of genes related to gibberellin (GA) and abscisic acid (ABA) pathways, and then negatively regulates the germination of seeds [27]; AtPIF4 promoting temperature-interceded blooming by directly activating the expression of *FLOWERING LOCUS T* (*FT*) [24], and AtPIF3 influences photomorphogenesis of plants by adversely controlling the expression of genes related to chlorophyll biosynthesis [28, 29]. In rice, OsPIL15 represses etiolated seedling growth; and integrates light and gravitropism signals to regulate tiller angle [30, 31]. OsPIL1 binds the promoters of the chlorophyll synthesizing genes *OsPORB* and *OsCAO1* to enhance their transcription, thereby enhancing chlorophyll synthesis and improving rice productivity [32]; Further studies showed that OsPIL1 negatively regulated rice leaf senescence by directly binding to the promoter region to upregulate the senescence inducing gene *ORE1* [33].

In the process of stresses, PIF regulates the expression of target genes by interacting with other proteins or binding the promoter DNA of downstream genes alone. For example, the expression of *AtPIF3* in *Arabidopsis thaliana* impacts plant cold tolerance and stomatal openness [34]. Specifically, AtPIF3 inhibits the expression of cold regulatory gene *CBFs* to alleviate plant growth inhibition caused by low temperature hypersensitivity [35]. AtPIF4 and AtPIF7 play a role in the thermomorphogenesis of *Arabidopsis*, AtPIF4 affects the rapid elongation of plant axis and leaf dysplasia under high temperature stress, AtPIF7 regulates the expression of temperature regulation genes in the early response to high temperature [36, 37]. *OsPIL13* overexpression restrains rice internode elongation under drought conditions by affecting the expression of cell wall related genes [38], and OsPIL16 negatively regulates rice cold resistance by affecting the expression of *OsDREB1* [39]. In maize, ZmPIF3 overexpression significantly enhances plant tolerance to drought and salt stress by affecting relative water content, chlorophyll content and chlorophyll fluorescence, stomatal closure, and enhanced cell membrane stability [40, 41]. Like to ZmPIF3, ZmPIF1 can also improve drought resistance in rice by inducing stomatal closure [42]. Although the function of PIF protein in some plants has been reported, systematic identification and functional studies of PIF have made PIF understudied in important agronomic crops such as bread wheat.

In this study, we identified and analyzed *PIF* genes in wheat at the genome-wide level. Firstly, the chromosomal location, gene structure, phylogenetic relationship, and protein physicochemical properties of TaPIFs were analyzed. Subsequently, the expression patterns of *TaDi19s* were analyzed by using publicly available transcriptomic data and quantitative reverse transcription PCR (qRT-PCR). Finally, transcriptional activation and subcellular localization of TaPIF2-1B and TaPIF4-5B were verified in yeast and wheat protoplasts, respectively. The results of this study provide new insights into the evolution and function of PIF in wheat and offer valuable information for future studies on PIF function.

## 2. Results

### 2.1. TaPIF identification, classification, and sequence analysis

PIF proteins were identified in the wheat genome based on sequence similarity to known AtPIFs. A total of 18 TaPIF proteins containing conserved bHLH and APB domains (Table 1 and S1 Fig). Thus, given the hexaploid nature of wheat with three subgenomes (2n = 6x = 42; AABBDD), we further classified the 18 TaPIFs into 6 homeologous groups. The similarity percentage of TaPIF proteins were 96.4% (TaPIF1s), 97.1% (TaPIF2s), 94.8% (TaPIF3s), 90.4% (TaPIF4s), 96.5% (TaPIF5s), 91.8% (TaPIF6s), respectively. TaPIF protein length ranged from 336 (TaPIF4-5A) to 516 (TaPIF5-5D) amino acids, with predicted molecular weight (MW) ranging from 36.6 kDa to 55.08 kDa. The predicted pI values varied from 5.37 (TaPIF4-5A) to 6.61 (TaPIF2-1B and TaPIF4-5B). In addition, the instability index of all the TaPIF proteins was greater than 53.56, which was much higher than 40, indicating that they were unstable; grand average of hydropathicity (GRAVY) analysis result shows that the GRAVY of TaPIF protein is all less than 0, ranging from -0.74 to -0.46, indicating that TaPIF proteins were a hydrophilic protein, and TaPIF1-1B is the most hydrophilic among them. Subcellular localization prediction suggested that all TaPIF proteins localized to the nucleus. In addition, we also predicted the above parameters of PIF protein in rice, *Arabidopsis*, *Brachypodium distachyon*, and maize (S1 Table), finding that these proteins were similar to TaPIF proteins on instability indexes, GRAVY scores, and subcellular localization. In particular, the hydrophilicity of PIF protein in *Arabidopsis* (-0.95 to -0.562) was generally higher than that in wheat.

### 2.2. Phylogenetic and syntenic analyses

To analyze the relationships between PIF family members, a phylogenetic tree was constructed with the 45 PIF proteins from wheat, *Arabidopsis*, *Z. mays*, *O. sativa*, and *B. distachyon* (Fig 1 and S2 Table). The PIF proteins from these five species were classified into four groups (I, II,

**Table 1. Characterization of PIFs in wheat.**

| Gene name | Locus_ID | Chr. | Position | | Deduced polypeptide | | | instability index | GRAVY | Subcellular localization |
|---|---|---|---|---|---|---|---|---|---|---|
| | | | Start | End | Length (aa) | MW (KDa) | pI | | | |
| *TaPIF1-1A* | TraesCS1A02G083000 | 1A | 66174370 | 66178373 | 507 | 54.41 | 6.31 | 79.06 | -0.53 | nucleus |
| *TaPIF1-1B* | TraesCS1B02G100400 | 1B | 109210350 | 109213975 | 390 | 42.14 | 6.53 | 83.48 | -0.74 | nucleus |
| *TaPIF1-1D* | TraesCS1D02G084200 | 1D | 67860728 | 67864379 | 502 | 53.89 | 6.37 | 75.51 | -0.50 | nucleus |
| *TaPIF2-1A* | TraesCS1A02G212700 | 1A | 375470020 | 375478638 | 455 | 48.42 | 5.88 | 55.07 | -0.47 | nucleus |
| *TaPIF2-1B* | TraesCS1B02G226200 | 1B | 406037910 | 406052114 | 454 | 48.37 | 6.61 | 56.43 | -0.51 | nucleus |
| *TaPIF2-1D* | TraesCS1D02G215600 | 1D | 300860435 | 300872558 | 454 | 48.18 | 5.87 | 53.56 | -0.46 | nucleus |
| *TaPIF3-2A* | TraesCS2A02G253900 | 2A | 386292343 | 386297900 | 347 | 36.86 | 6.05 | 69.09 | -0.56 | nucleus |
| *TaPIF3-2B* | TraesCS2B02G273500 | 2B | 374801132 | 374803415 | 358 | 38.31 | 5.77 | 67.10 | -0.59 | nucleus |
| *TaPIF3-2D* | TraesCS2D02G254400 | 2D | 306528220 | 306533600 | 350 | 37.46 | 5.79 | 64.95 | -0.57 | nucleus |
| *TaPIF4-5A* | TraesCS5A02G049600 | 5A | 45369032 | 45372490 | 336 | 36.60 | 5.37 | 63.31 | -0.63 | nucleus |
| *TaPIF4-5B* | TraesCS5B02G054800 | 5B | 59925570 | 59928669 | 338 | 36.87 | 6.61 | 66.58 | -0.52 | nucleus |
| *TaPIF4-5D* | TraesCS5D02G060300 | 5D | 56950525 | 56953479 | 350 | 37.96 | 5.87 | 62.38 | -0.505 | nucleus |
| *TaPIF5-5A* | TraesCS5A02G376500 | 5A | 574193926 | 574200864 | 487 | 51.69 | 6.39 | 65.95 | -0.59 | nucleus |
| *TaPIF5-5B* | TraesCS5B02G380200 | 5B | 558117488 | 558124468 | 447 | 47.69 | 6.08 | 61.79 | -0.58 | nucleus |
| *TaPIF5-5D* | TraesCS5D02G386500 | 5D | 456338580 | 456345166 | 516 | 55.09 | 6.40 | 64.87 | -0.58 | nucleus |
| *TaPIF6-5A* | TraesCS5A02G420200 | 5A | 606892572 | 606895950 | 420 | 44.61 | 6.53 | 70.81 | -0.55 | nucleus |
| *TaPIF6-5B* | TraesCS5B02G422000 | 5B | 597217607 | 597223578 | 410 | 43.24 | 6.50 | 71.11 | -0.57 | nucleus |
| *TaPIF6-5B* | TraesCS5B02G422000 | 5B | 597217607 | 597223578 | 410 | 43.24 | 6.50 | 71.11 | -0.57 | nucleus |

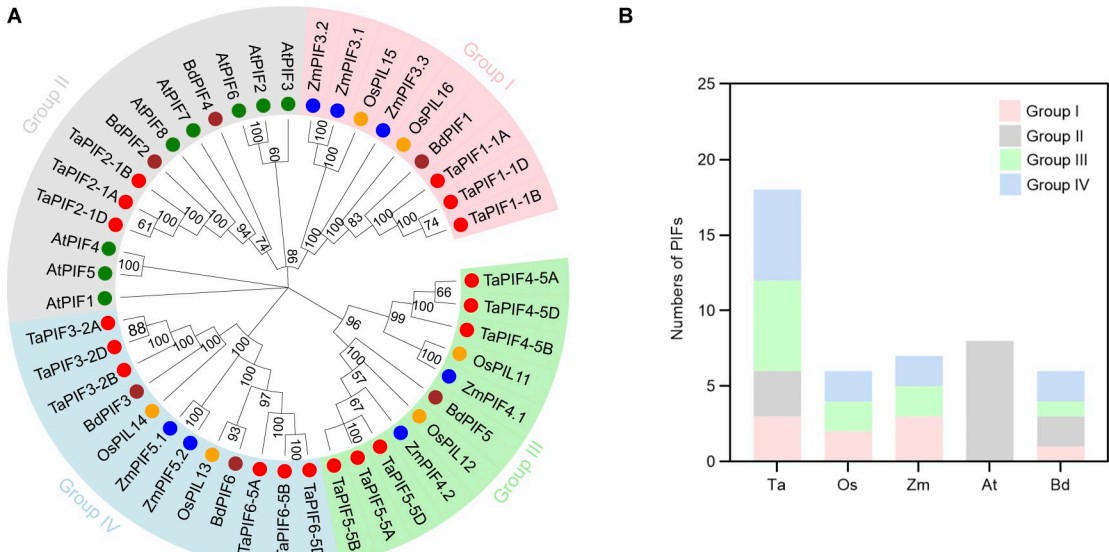

**Fig 1. Phylogenetic analysis of PIF families in *Arabidopsis thaliana*, rice, maize, *Brachypodium distachyon*, and wheat.** (A) Relationships between PIFs in wheat, *Oryza sativa*, *Arabidopsis*, *B. distachyon*, and *Zea mays*. The neighbor-joining phylogenetic tree was constructed using the full-length protein sequences of all known PIFs in each species. The proteins could be classified into five groups (I–V). The bootstrap scores shown at each node were calculated from 1000 replicates. (B) Distribution of PIFs among each group in each of the five species.

III, and IV) (Fig 1). TaPIFs and BdPIFs were distributed among all four groups, ZmPIFs and OsPIFs were clustered in group I, group III, and group IV. Furthermore, all AtPIFs were aggregated in group II, while OsPIF proteins were distributed in all groups except group II. TaPIF1-1A/B/D clustered with BdPIF1, OsPIL15, OsPIL16, ZmPIF3.1, ZmPIF3.2, and ZmPIF3.3 in group I; TaPIF2-1A/B/D clustered with BdPIF2, BdPIF3 and all AtPIFs in group II; and TaPIF4-5A/B/D and TaPIF5-5A/B/D clustered with BdPIF5, ZmPIF4.1, ZmPIF4.2, OsPIL11, and OsPIL12 in group III, the remaining proteins cluster in group IV (Fig 1A).

TaPIF genes that were identified could be clustered into 6 homoeologous groups in wheat. Most duplication events in the wheat genome have been between copies of a single chromosome in different subgenomes, such as *TaPIF1-1A*, *TaPIF1-1B*, and *TaPIF1-1D*; while there have also been inter-chromosomal gene replication events, such as those between chromosome 2 and chromosome 5 (Fig 2A and S3 Table), illustrated by a collinearity between *TaPIF3-2A/D* on chromosomes 2 and *TaPIF6-5A/B/D* on chromosomes 5. These duplication events resulted in 23 collinear pairs of *TaPIFs* in the wheat genome, of which 18 collinear pairs were due to genome-wide duplication (WGD) (Fig 2A and S3 Table).

Further, we performed collinearity analysis between wheat and *O. sativa* and *B. distachyon* to understand the evolutionary relationships between members of the PIFs. There were 24, and 25 collinear *PIF* gene pairs identified in wheat/*O. sativa* and wheat/*B. distachyon*, respectively (Fig 2B and S3 Table). Notably, chromosomes 5A/B/D of wheat exhibited the highest homology with *O. sativa* and *B. distachyon*. *PIF* genes had been subject to strong purifying selection in rice and *B. distachyon* as their Ka/Ks ratios are less than 1 [43], and the similar number of gene duplication events was found in wheat/*O. sativa* (24) and wheat/*B. distachyon* (25), respectively. Specifically, *TaPIF1-1A/B/D* were collinear with *OsPIL16*; *TaPIF3-2A/B/D* were collinear with *OsPIL13* and *OsPIL14*; *TaPIF4-5A/B/D* were collinear with *OsPIL11*; *TaPIF5-5A/B/D* were collinear with *OsPIL11* and *OsPIL13*; *TaPIF6-5A/B/D* were collinear with *OsPIL13* and *OsPIL14*. There were as many as 25 gene pairs between *B. distachyon* and

**A**

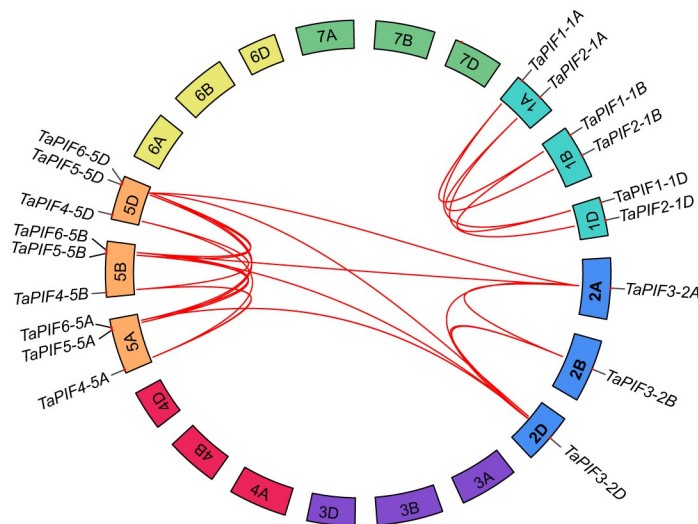

**B**

**Fig 2. *PIF* synteny analysis.** (A) Synteny analysis of *PIFs* in the wheat genome. Isomorphic *TaPIF* gene pairs are connected with red lines. (B) Synteny analysis of *Di19* genes in wheat, *Oryza sativa*, *Arabidopsis thaliana*, and *Brachypodium distachyon*. The orange bars represent chromosomes in wheat, while the green bars represent chromosomes in rice and B. distachyon. The text on the sides of the bars indicates the chromosome numbers. Gray lines indicate collinearity between the wheat genome and those of other plant species. Red lines show isomorphic *PIF* gene pairs.

wheat, for example *TaPIF1-1A/B/D*, *TaPIF1-1A/B/D*, *TaPIF2-1A/B/D*, and *TaPIF3-2A/B/D* collinear with *BdPIF1*, *BdPIF2*, and *BdPIF3* (S2 Table).

### 2.3. *TaPIF* gene structure and conserved protein motifs

Investigating structural diversity among *TaPIF* genes, a phylogenetic tree was constructed using only the 18 full-length TaPIF protein sequences (Fig 2). We mapped the exon-intron organization in the coding regions of *TaPIF* genes, revealing that each *TaPIF* gene contains four to seven exons. Specifically, *TaPIF1*, *TaPIF4*, and *TaPIF5* had four exons; *TaPIF2* has five exons; *TaPIF3* and *TaPIF7* had seven exons (Fig 3).

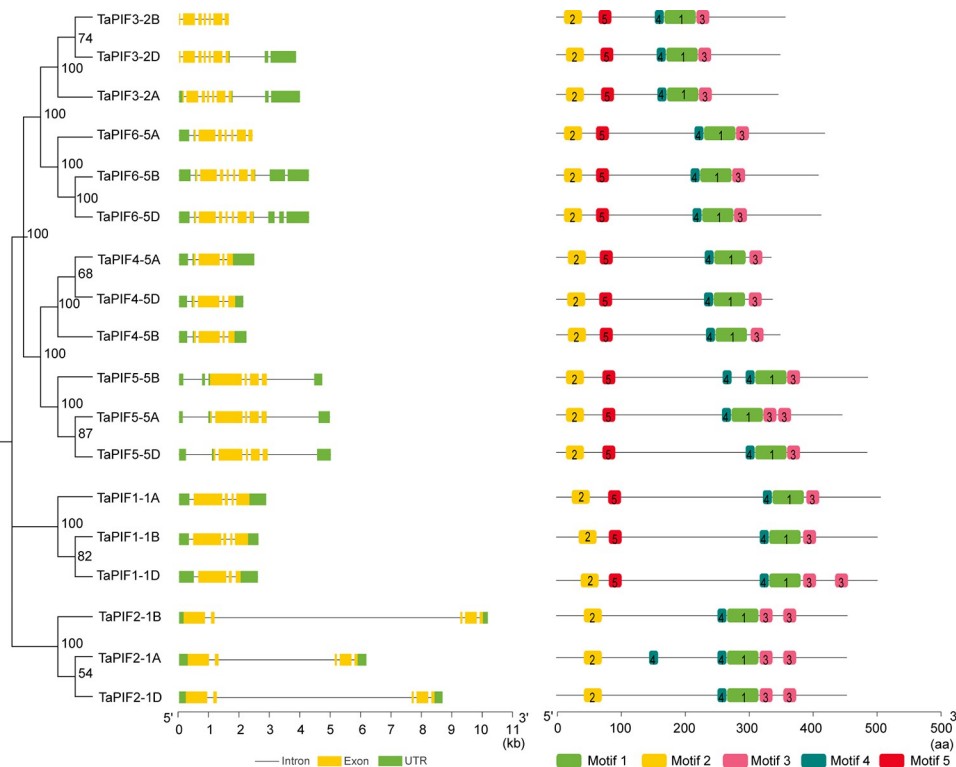

**Fig 3. Phylogenetic relationships, conserved motifs, and gene structures of TaPIFs.** Phylogenetic analysis of PIFs in wheat and four other plant species clustered the proteins into five groups. Left, conserved motifs among the TaPIFs. Right, gene structures of *TaPIF*s. Untranslated regions are indicated with green boxes, exons are indicated with yellow boxes, and introns are represented by black lines.

To further characterize structural diversity and determine the functions of TaPIFs, the MEME suite software was used to identify 5 motifs among the TaPIFs (Fig 3 and S4 Table and S2 Fig). The most closely-related family members (as determined from the phylogenetic analysis) showed similar motif arrangements and positions, indicating that PIFs within each group may performed similar biological functions. All of the TaPIFs contained the same arrangement of motif 1 and motif 2, representing bHLH and APB domains, the features of PIF proteins, respectively. The conservation of some motifs between groups suggested that these proteins may all have similar functions, although the presence of group-specific domains and motifs would allow for functional specialization.

## 2.4. *Cis*-acting element analysis

Gene transcription is regulated by *cis*-acting elements that function as TF binding sites [44]. Upstream regulatory elements analysis can provide clues to the potential functions of target genes. Here, we conducted *cis*-element analysis in the 2-kb region upstream of the translation start site (TSS) in *TaPIF*s, Revealing the presence of *cis*-elements involved in growth and development, stress responses, and hormone responses (Fig 4 and S5 Table). The *TaPIF* promoters contained many hormone-related motifs, including auxin response elements (TGA-elements and AuxRE-cores), GA response elements (P-boxes, GARE-motifs, and TATC-boxes), ABA response elements (ABREs), SA response elements (TCA-elements), and methyl jasmonate (MeJA) response elements (TGACG-motifs and CGTCA-motifs). ABA- and MeJA-response

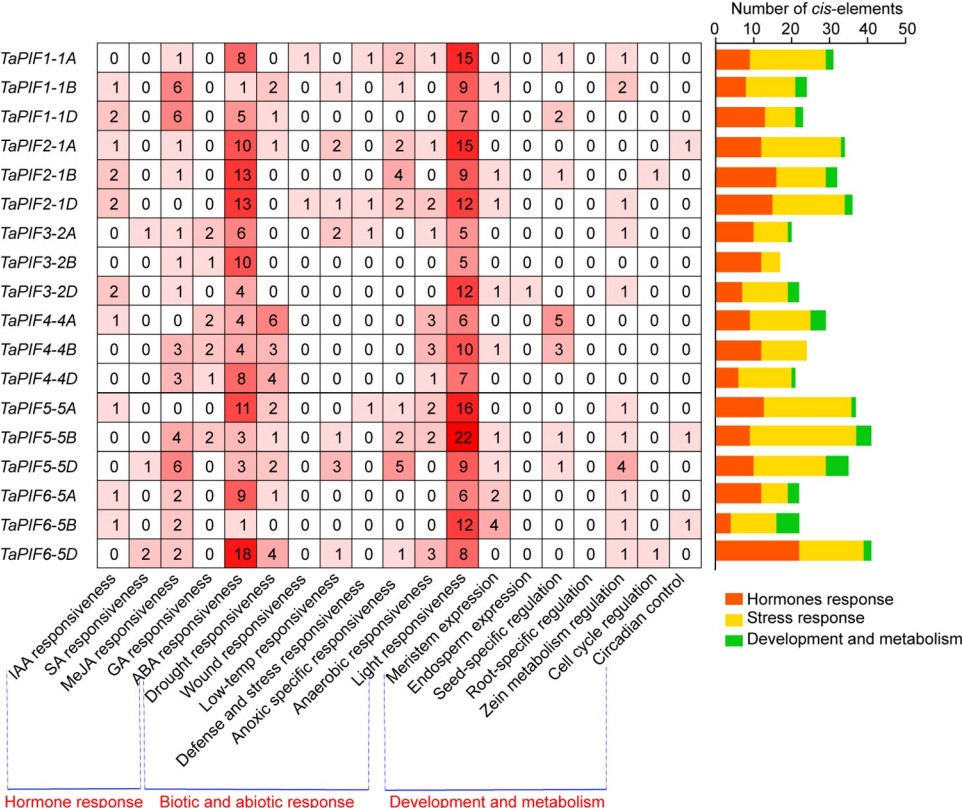

**Fig 4. Analysis of *TaPIF* promoter *cis*-elements.** Analysis of *TaPIF* promoter *cis*-elements. The 2-kb promoter sequences of *TaPIF*s were analyzed to identify predicted *cis*-elements. The identified *cis*-elements were classified into three categories: hormone-related, development-related, and abiotic stress-related. The color intensity and number in each square indicate the number of each type of *cis*-element in the promoter region of the indicated gene. The distribution of *cis*-elements in each category for each gene is shown at right.

elements were the most abundant hormone-related *cis*-elements, present in the promoters of all *TaPIF*.

A large number of stress-related *cis*-elements were also found in *TaPIFs* promoters. Light-response elements (G-boxes, TCT-motifs, I-boxes, Sp1s, and MREs) were found in the promoters of all *TaPIF* genes and were the most abundant class of *cis*-elements. There were 11 *TaPIFs* containing drought-inducible MYB binding sites (MBSs) or dehydration-responsive elements (DREs) in promoters, including *TaPIF1-1A/B/D*, *TaPIF4-5A/B/D*, *TaPIF5-5A/B/D*, and *TaPIF6-5A/D*, suggesting that these genes may be involved in wheat drought responses. Low-temperature responsive (LTR) elements, another major type of stress-related motif, were found in the promoters of *TaPIF1-1B*, *TaPIF2-1A/D*, *TaPIF3-2A*, *TaPIF5-5B/D*, and *TaPIF6-5D*. The promoter of *TaPIF1-1A* and *TaPIF2-1D* contained the wound-responsive WUN-motif, and four *TaPIFs* contained TC-rich repeats, which are involved in defense and stress responses. Notably, some of the identified *cis*-elements were involved in organ- or tissue-specific processes, such as endosperm expression (GCN4 motif), meristem expression (CAT-box), root-specific expression (motif I), seed-specific regulation (RY-element), cell cycle regulation (MSA-like), and zein metabolism regulation (O2-site) (Fig 4 and S5 Table). The promoter of *TaPIF3-2D* contained endosperm expression and zein metabolism regulation elements, suggesting that it may play key roles in wheat seed development. These results demonstrated that *TaPIF* genes may be involved in plant development and in hormone and stress responses.

## 2.5. Tissue-specific *TaPIF* expression

RNA-seq is an effective method for studying gene expression patterns with high throughput. Using RNA-seq data from public databases to obtain specific gene expression levels is one way to predict gene function [45]. To investigate the potential biological functions of *TaPIFs*, RNA-seq data were analyzed to identify *TaPIF* expression patterns during wheat growth and development, including roots, stems, leaves, flag leaves, spikes, and grains. A systematic clustering analysis was performed for all 18 *TaPIFs* based on their spatiotemporal expression patterns (i.e., expression in various tissues and at different developmental timepoints) (Fig 5 and S5 Table). Generally, *TaPIFs* were expressed in many tissue types across multiple developmental stages. *TaPIFs* were expressed at particularly high levels in the stems, leaves, flag leaves, and young spikes. Specifically, *TaPIF1s* had high expression in stem, flag leaves, and young spikes; *TaPIF2s* had high expression in leaves, flag leaves, and young spikes; *TaPIF3s*, *TaPIF4s*, and *TaPIF5s* were highly expressed in the stem, leaves, flag leaves, and young spikes; *TaPIF6s* were most highly expressed in leaves and flag leaves. These expression patterns indicating that *TaPIFs* likely have strongly influential functions in wheat development.

## 2.6. *TaPIF* responses to biotic stress

To understand the roles of *PIFs* under biotic stress conditions, we analyzed the expression patterns of *TaPIFs* in wheat infected with stripe rust or powdery mildew (Fig 6 and S6 Table). *TaPIF1-1A/B/D*, *TaPIF5-5A/B/D*, and *TaPIF6-5A/D* were significantly up-regulated at 24 h of treatment with stripe rust, while *TaPIF2-1A/B/D* were strongly down-regulated at 24 h after stripe rust infection. There were no significant changes in expression levels of *TaPIF3A/B/D*

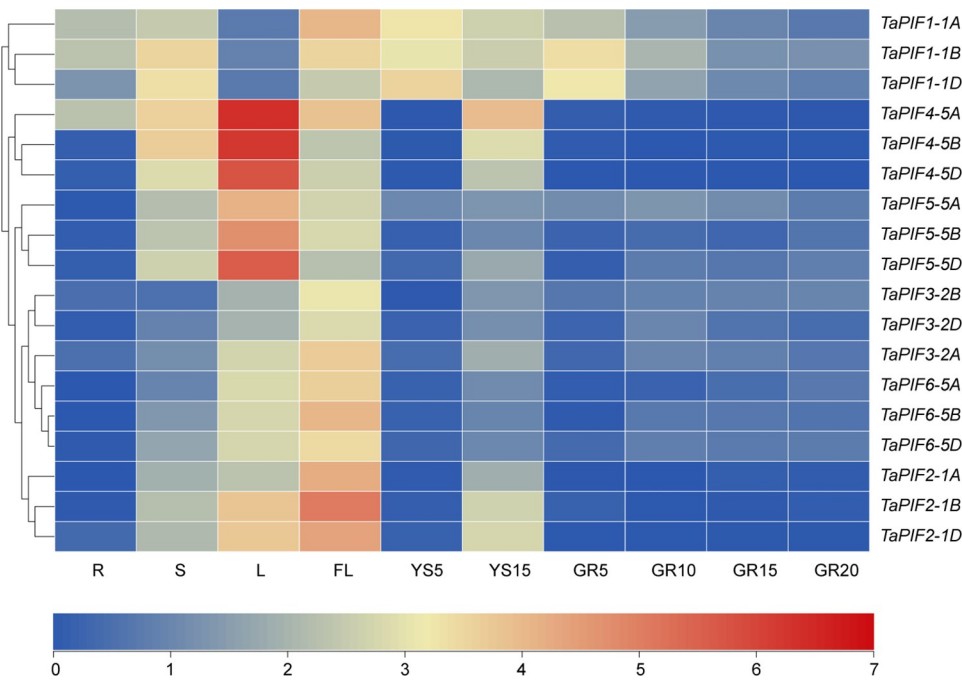

**Fig 5. Tissue-specific *TaPIF* expression patterns.** Expression profiles of *TaPIF* genes across multiple tissue types. Analyses were conducted using publicly-available RNA-seq data. R, S, and L indicate the root, stem, and leaf tissue, respectively, from seedlings at the five-leaf stage. FL refers to the flag leaf of wheat plants at the booting and heading stages. YS5 and YS15 indicate young spikes from wheat plants at the booting and heading stages, respectively. GR5, GR10, GR15, and GR20 refer to grains at 5, 10, 15, and 20 d post-fertilization, respectively. All expression values were log₂ transformed.

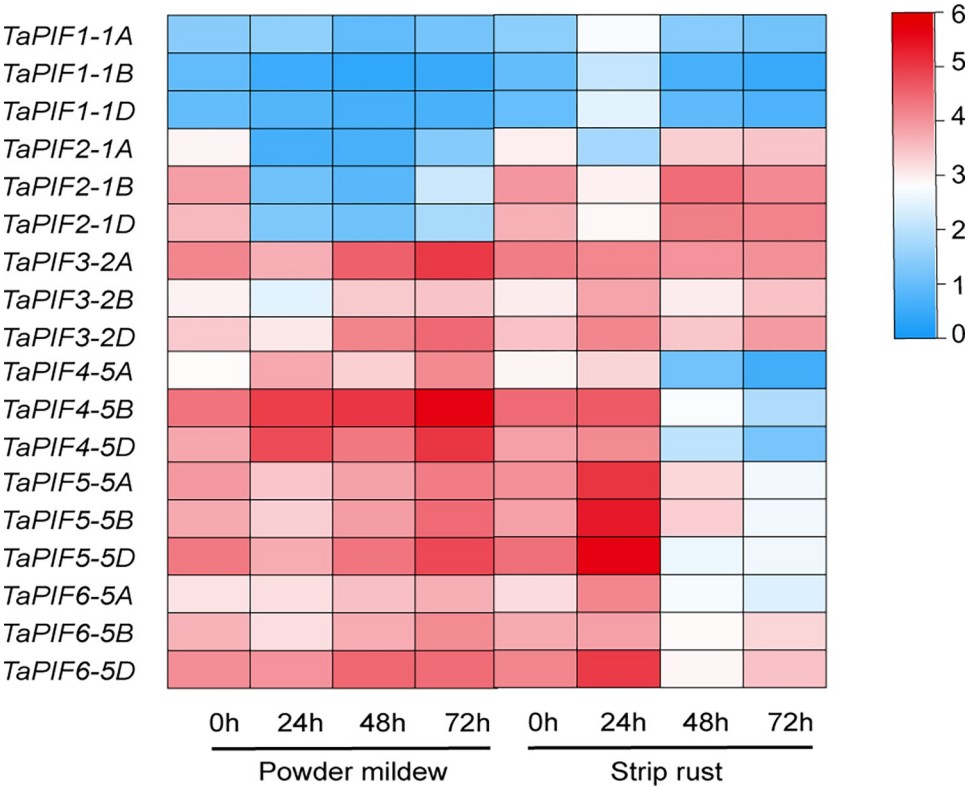

**Fig 6. *TaPIF* expression in response to biotic stress.** Expression profiles of *TaPIF* genes in wheat after infection with powdery mildew or stripe rust. Analyses were conducted using publicly-available RNA-seq data. All expression values were log$_2$ transformed. The color bars represent the level of gene expression, with blue indicating low expression and red indicating high expression.

and *TaPIF4A/B/D* between 0 and 72 h. *TaPIF4-5A/B/D* were up-regulated in response to powdery mildew treatment, whereas *TaPIF2-1A/B/D*, *TaPIF3-2A/B/D*, and *TaPIF5-5A/B/D* were down-regulated. There were no significant differences in expression levels of *TaPIF1-1A/D* and *TaPIF6-5A/B/D* (Fig 6 and S6 Table). Combined with the above *cis*-acting element identification, the promoter regions of *TaPIF1-1A/B/D*, *TaPIF2-1A/B/D*, *TaPIF5-5A/B/D*, and *TaPIF6-5A/B/D* identified defense and stress responses, SA response, or MeJA response elements (Fig 4 and S5 Table), which may be the reason for the response of these genes to the infection of stripe rust or powdery mildew. These results indicating both similarities in stress responses among some *TaPIFs* and diversity in expression profiles between other family members.

## 2.7. *TaPIF* responses to abiotic stress and hormone treatment

To explore potential functions of *PIFs* in abiotic stress responses, weanalyzed the expression patterns of *TaPIFs* in wheat treated with drought, heat, cold, and salt stress treatments (Fig 7A and 7B and S6 Table). Under drought conditions, *TaPIF1-1A/B/D* were up-regulated compared with control plants; there were no significant changes in *TaPIF4* expression levels. The remainder of TaPIF genes were down-regulated, consistent with their promoter region containing drought-response elements like MBS or DRE (Figs 4 and 7A and S6 Table). Under heat stress, *TaPIF6-5A/D* were up-regulated, *TaPIF3-2A/B/D* and *TaPIF5-5A/B/D* were down-regulated at 1 h and 6 h treatments, respectively (Fig 7A and S6 Table). Under salt stress, *TaPIF3-*

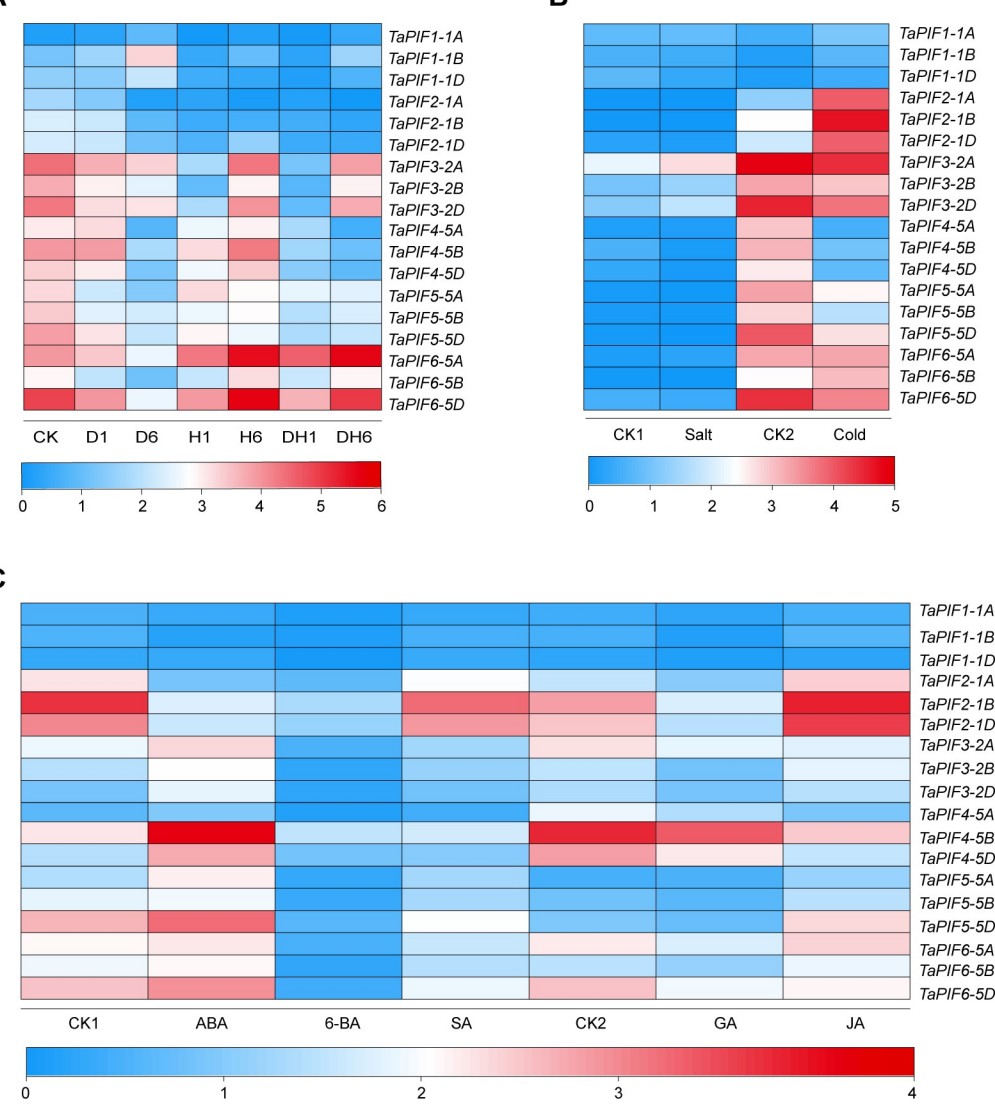

**Fig 7. Gene expression patterns of *TaPIF*s in response to abiotic stress or phytohormone treatment.** (A) Expression profiles of *TaPIF* genes in wheat exposed to drought (D), heat (H), or combined drought and heat (D&H). (B) Expression profiles of *TaPIF* genes in wheat exposed to salt or cold stress. CK1 and CK2 were the control samples for the salt and cold stress treatments, respectively. (C) Expression profiles of *TaPIF* genes in wheat exposed to abscisic acid (ABA), 6-benzylaminopurine (6-BA), salicylic acid (SA), gibberellic acid (GA), or jasmonic acid (JA). CK1 was the control for the ABA, 6-BA, and SA treatments; CK2 was the control for the GA and JA treatments. Analyses were conducted using publicly-available RNA-seq data. All expression values were log$_2$ transformed. The color bars represent the level of gene expression, with blue indicating low expression and red indicating high expression.

*2A/D* and *TaPIF6-5A* were up-regulated, and *TaPIF1-1D*, *TaPIF4-5B/D*, and *TaPIF5-5A/B/D* were down-regulated. Besides, *TaPIF1-1A/B/D*, *TaPIF2-1A/B/D*, and *TaPIF6-5D* were up-regulated under cold conditions, whereas *TaPIF3-2D*, *TaPIF4-5B/D*, and *TaPIF5-5A/B/D* were down-regulated (Fig 7B and S6 Table), which may be due to the existence of LTR element in promoter region of *TaPIF1-1B*, *TaPIF2-2A/D*, *TaPIF3-2A*, *TaPIF5-5B/D*, *TaPIF6-5D* (Fig 4). These results again showed both commonalities and diversity in *TaPIF* responses to stressors.

Plant hormones are involved in all aspects of plant development. We therefore analyzed PIF expression patterns in response to treatment with the phytohormones ABA, 6-BA, SA,

GA, and JA ([Fig 7C] and [S6 Table]). *TaPIF3-2A/B*, *TaPIF4-5A/B/D*, *TaPIF5-5A/B/D*, and *TaPIF6-5A/B/D* were down-regulated, while *TaPIF2-1B/D* were up-regulated in response to SA treatment, despite the fact that only *TaPIF3-2A*, *TaPIF5-5D*, and *TaPIF6-5D* had SA-responsive elements in their promoters. The expression of all *TaPIFs* were inhibited by 6-BA treatment; *TaPIF1-1A/B*, *TaPIF2-1A/B/D* were down-regulated, while *TaPIF3-2A/B/D*, *TaPIF4-5A/B/D*, and *TaPIF5-5A/B/D* were up-regulated in response to ABA exposure compared to control, which were inseparable from the ABRE-element of their promoters ([Fig 7C] and [S6 Table]). For GA and JA treatments, except that the expression level of *TaPIF5-5A/B/D* did not show significance, the remaining *TaPIF* genes were down-regulated under GA treatments; *TaPIF2-1A/B/D*, *TaPIF4-5A/B/D*, and *TaPIF5-5A/B/D* had obvious response to exposure to JA ([Fig 7C] and [S6 Table]). Overall, *TaPIFs* had divergent responses to ABA, 6-BA, SA, GA, and JA, suggesting that the *TaPIFs* differed in hormonal pathway membership.

## 2.8. *TaPIF* responses to PEG and ABA treatment

To investigate if *TaPIF* genes were drought-responsive, we used qRT-PCR to examine *TaPIF* expression in the leaves of plants treated with PEG-induced osmotic stress. Because ABA is a signal transducer in the drought stress response pathway, we also analyzed the effects of ABA treatment on *TaPIF* expression. Allele pairs from the A-, B-, and D-subgenomes were grouped and tested together due to the high sequence similarity in the corresponding transcripts. The analyzed genes exhibited diverse responses to the drought stress treatments ([Fig 8]). *TaPIF1* and *TaPIF2* were up-regulated in response to PEG treatment, especially at 12 h, indicating the high responsiveness of these two genes to osmotic stress. *TaPIF3* and *TaPIF5* showed highly similar expression profiles in response to osmotic stress. *TaPIF6s* were down-regulated to similar degrees. At 1 h after ABA treatment, *TaPIF1s*, *TaPIF2s*, and *TaPIF6s* were significantly up-regulated, and *TaPIF3s* and *TaPIF5s* were up-regulated at 6 h. These expression patterns indicate that *TaPIFs* may play an important role in osmotic stress by ABA dependent pathway.

## 2.9. TaPIF transactivation activity and subcellular localization

We next sought to isolate and functionally characterize several *TaPIFs*. Transactivation assays were performed in yeast with TaPIF2 and TaPIF4. Due to the high similarity among homologous genes, one representative homolog was selected for experimental validation. *TaPIF2-1D* and *TaPIF4-5A* were cloned from the wheat cultivar 'Chinese Spring', and inserted into the vector pGBKT7, and transformed into yeast. TaPIF2-1B and TaPIF4-5B promoted increased yeast growth on SD/-Trp/-His/-Ade medium compared to yeast transformed with the empty vector negative control ([Fig 9A]). The α-galactosidase activity confirmed these results. These results revealed that TaPIF2 and TaPIF4 had transactivation activity.

Sequence analysis predicted that each of the TaPIFs would be localized to the nucleus ([Table 1]). To confirm these subcellular localization predictions, we developed transient expression vectors to express plasmids encoding TaPIF-GFP fusion proteins (TaPIF2-1B-GFP and TaPIF4-5B-GFP) in wheat protoplasts. As expected, visualization of transformed cells showed that TaPIF2 and TaPIF4 proteins were localized to the nucleus ([Fig 9B]).

## 3. Discussion

### 3.1. Identification and evolutionary analysis of the PIF family

PIF proteins comprise a subfamily of the bHLH TF superfamily, and functional characterizations have shown that PIF proteins are involved in plant growth, development, and stress responses [14, 26, 46]. In this study, we identified 18 *PIF* genes in wheat, categorized into 6

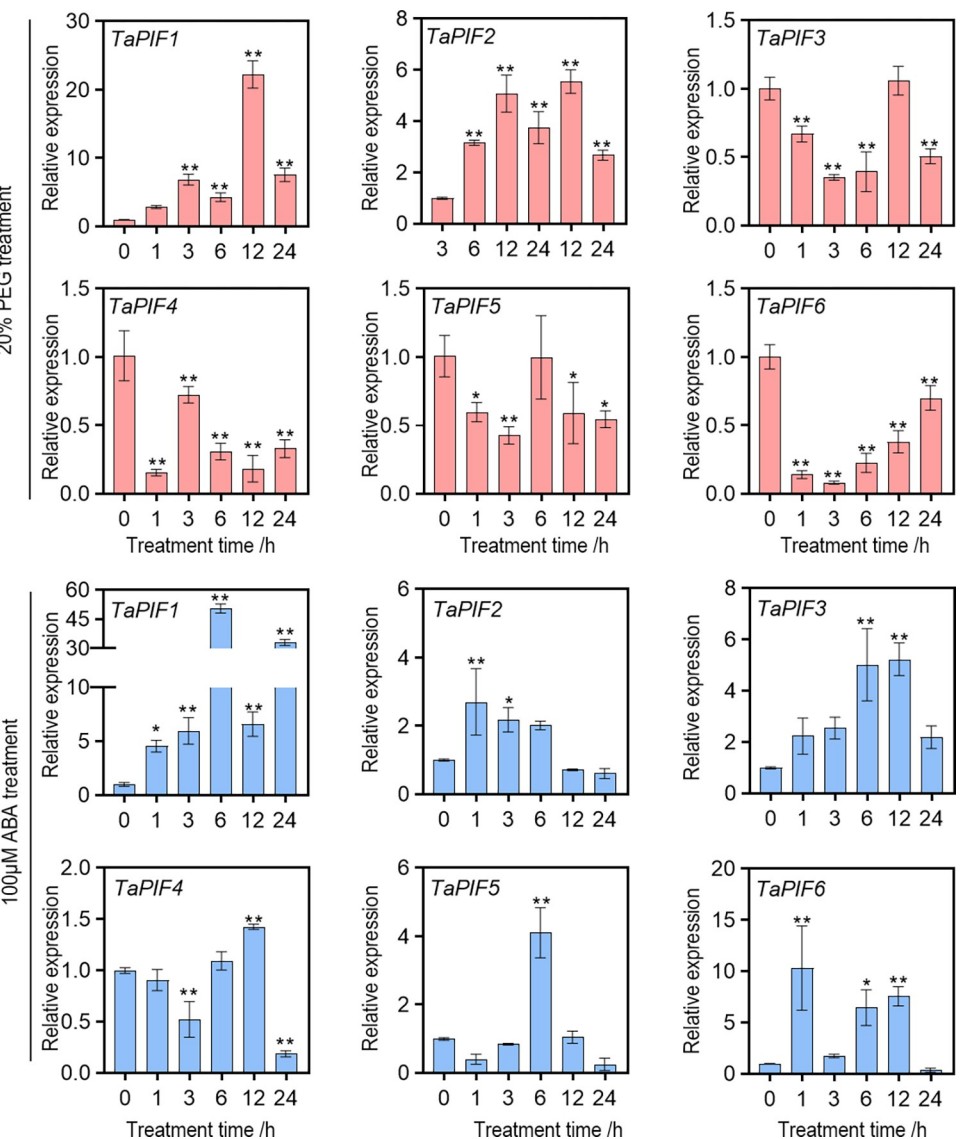

**Fig 8. Expression profiles of *TaPIFs* treated with polyethylene glycol (PEG) or abscisic acid (ABA).** *TaPIF* expression in wheat shoots and roots in response to treatment with 20% PEG-6000 or 100 μM ABA. Expression was measured with quantitative reverse transcription PCR. Data are presented as the mean ± standard deviation ($n$ = 3). The asterisks indicate significant differences based on the results of the one-way ANOVA test (* $P < 0.05$; ** $P < 0.01$).

homeologous groups (Table 1), which is a greater number of *PIF* genes than in the *Arabidopsis*, rice, maize, or *B. distachyon genomes* (S1 Table). This increase is likely due to wheat being a heterohexaploid with three subgenomes. Polyploid events or whole genome duplication (WGD) and amplification events play an important role in the genome evolution of plant species [47]. Previous studies have shown that the expansion of gene families is mostly driven by gene duplication events such as WGD, segmental duplication, and tandem duplication events [48]. Here, collinearity analysis of wheat/wheat, wheat/*O. sativa*, and wheat/*B. distachyon* revealed 21, 24, and 25 collinearity *PIF* gene pairs, respectively (Fig 2 and S2 Table). In the monocot species rice and *B. distachyon*, *PIF* genes showed evidence of purifying selection. Additionally, there had no collinearity gene pairs between *Arabidopsis* and wheat, it may be due to differences between monocots and dicots (S2 Table).

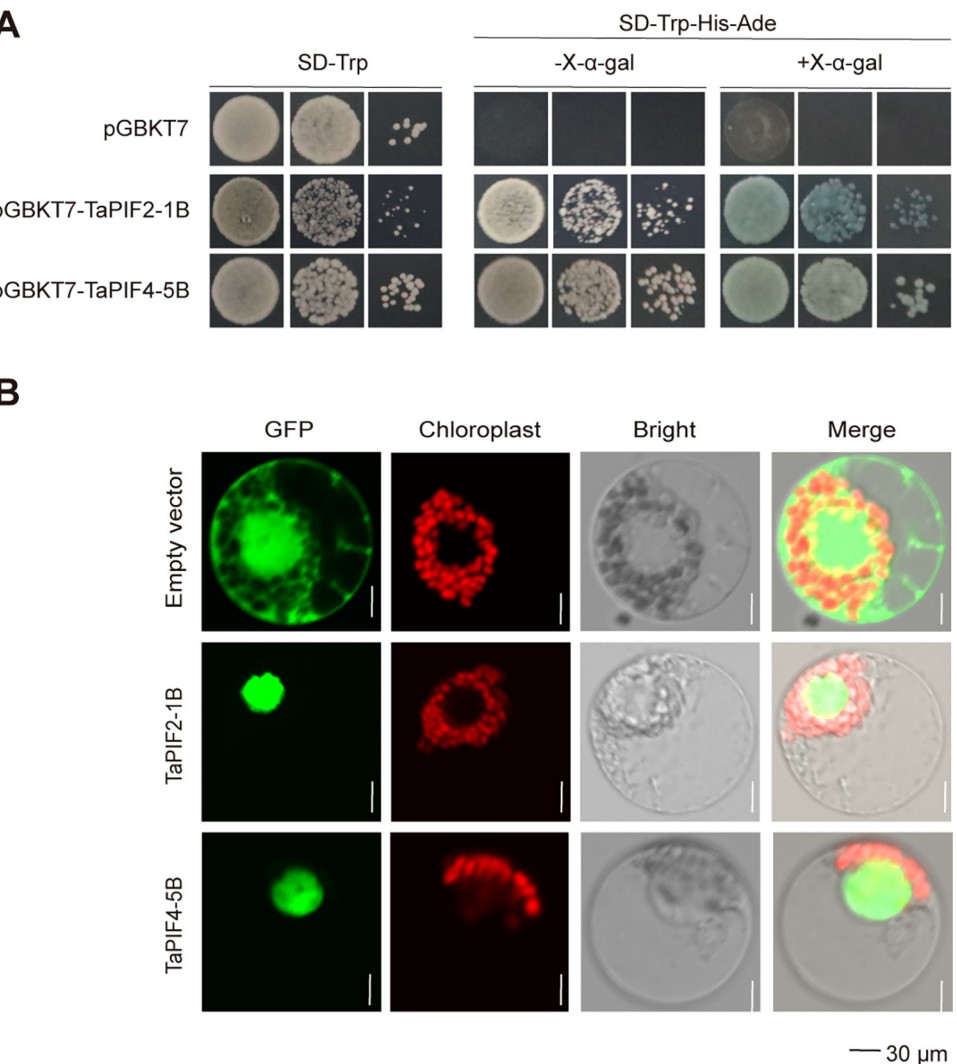

**Fig 9. Subcellular localization and transactivation activity of TaPIFs in wheat.** (A) Transactivation activity analyses of five TaPIFs in yeast. The tested proteins were TaPIF2 and TaPIF4. Transformed yeast cells were grown on the synthetic media. (B) Subcellular localization of TaPIFs in wheat mesophyll protoplasts. Protoplasts were transfected with 35S::*TaPIF2-GFP* and 35S::*TaPIF4-GFP*, or the 35S::*GFP* control and visualized with laser scanning confocal microscopy. Scale bar = 30 μm.

Phylogenetic analysis showed that PIF proteins in five species could be divided into four subgroups (Fig 1A), most PIF proteins in monocots (wheat, rice, maize, and *B. distachyon*) were distributed among groups I, III and IV, whereas all of the PIFs in the dicots (*Arabidopsis*) were distributed in groups II (Fig 1B). These results indicate that there were differences in PIF sequences between monocotyledonous and dicotyledonous plants (Fig 1). Phylogenetic analysis, combined with gene structure and protein motif analysis, showed that TaPIFs in each sub-clade had a similar protein length, gene structure and motif composition. This implies a high degree of evolutionary conservation among TaPIFs (Fig 3). Additionally, we have observed that the APB domain and bHLH domain tend to be located at the N-terminus and C-terminus in AtPIF proteins, respectively (S2 Fig). This similar arrangement pattern is also observed in all TaPIFs proteins, which strongly supports the claim that the identified TaPIF proteins were functionally similar to AtPIF proteins [7].

## 3.2. *TaPIF* expression was tissue-specific

The expression of *PIF* is largely tissue-specific and strongly influences core developmental characteristics in plants. For example, *OsPIL13* was reported to be highly expressed in the stem segment, which affects internode elongation and thus changes the height of the plant [37]. *ZmPIF1* and *ZmPIF3* were highly expressed in the pistils and leaves in maize [16, 40]. In tomato, *SlPIF4* was highly expressed in the leaves and pre-ripening fruits, and significantly down-regulated after ripening [49]. Our study found that most *TaPIFs* genes expressions were highly in leaves, flag leaves, and spikes, while low in grains (Fig 5). Besides, TaPIF1-1A/B/D, TaPIF4-5A/B/D, and TaPIF5-5A/B/D were also expressed at higher levels in the stem (Fig 5). The spatiotemporal differences in the expression of these genes may be related to wheat growth and developmental processes.

## 3.3. TaPIFs played important roles in hormone signaling pathways

Many *PIFs* have been reported to participate in multiple hormone signaling pathways in *Arabidopsis* [50], such as *AtPIF1*'s involvement in the regulation of ABA and GA signaling pathways [51, 52], and *AtPIF4* and *AtPIF5* participating in the auxin signaling pathway [53]. In this study, we found that most wheat *PIF* genes contained at least one hormone-responsive *cis*-element in the promoter region (Fig 4 and S5 Table). *TaPIF2-1A/B/D* were down-regulated in response to treatment with either ABA or 6-BA; *TaPIF1-1A/B/D*, *TaPIF3-2A/B/D*, and *TaPIF6-5A/B/D*, and *TaPIF10-6A/B/D* were down-regulated by 6-BA treatment; and *TaPIF3-2A/B/D*, *TaPIF4-5A/B/D*, and *TaPIF5-5A/B/D* were up-regulated by ABA (Fig 7C and S6 Table). Notably, some of the homologous *PIF* gene pairs showed differing expression patterns in response to ABA, SA, JA, or GA treatment. For example, *TaPIF3-2A* and *TaPIF6-5D* were down-regulated in response to JA, whereas *TaPIF3-2B* and *TaPIF6-5B* were up-regulated (Fig 7C and S5 Table); the similar phenomenon occurs between *TaPIF5-5A/D* and *TaPIF5-5B* treated with ABA. These results suggested that *PIFs* may be involved in the regulatory network governing responses to numerous hormones in wheat, thereby influencing growth and defense responses.

## 3.4. *TaPIFs* responded to multiple abiotic stresses

*Cis*-acting elements of gene promoter region play a crucial role in regulating gene expression by recruiting specific transcription factors [54–58]. In the promoter regions of TaPIF genes, we identified cis-elements related to three main categories: development and metabolism, hormone responses, and stress responses (Fig 4). These findings underscore the pivotal roles of *TaPIF* genes in both plant development and stress responses. Furthermore, gene expression analysis revealed diversified expression patterns among *TaPIFs*, with specific genes exhibiting unique spatiotemporal expression profiles (Figs 5–7). These varied expression patterns highlight the functional diversification of *TaPIFs* in contributing to wheat growth and development.

Notably, *TaPIF2*, *TaPIF3*, *TaPIF4*, *TaPIF5*, and *TaPIF6* exhibited significant differential expression in response to powdery mildew and stripe rust infection. Specifically, *TaPIF4-5A/B/D* showed up-regulated after stripe rust infection and down-regulated in response to powdery mildew, a pattern similarly observed in *TaPIF5* and *TaPIF6*. Pathogen defense is a complex process involving phytohormone crosstalk, with SA and JA playing major roles [59–61]. Intriguingly, *TaPIF2*, *TaPIF3*, *TaPIF4*, *TaPIF5*, and *TaPIF6* demonstrated distinct responses to SA and JA (Fig 6), aligning with the expression profiles of known pathogen-responsive genes. Moreover, *TaPIF1-1A/B/D* expression was induced by both cold and drought stresses, while *TaPIF6-5A* responded to both salt and cold stress. Furthermore, all 18 *TaPIFs* were differentially

expressed (to varying degrees) in response to PEG-induced simulated drought or ABA treatment (Fig 8). The functionalities suggested by the expression profiles closely resemble the known functions of *PIFs* in other crop species, such as rice and maize [24, 40, 62]. These results collectively provide a valuable reference for future validations of *TaPIF* functions.

### 3.5. TaPIFs functioned in the nucleus

In vitro transcriptional activity assays conducted with TaPIF2-1B and TaPIF4-5B provided empirical evidence supporting the role of TaPIFs serve as transcriptional activators (Fig 9A). Subsequent subcellular localization assays confirmed the nuclear localization of TaPIF2-1B and TaPIF4-5B, as predicted (Fig 9B), consistent with the cellular compartment of PIF proteins in *Arabidopsis* [63–65], rice [38, 39, 66], and maize [40, 41]. During plant growth, development and stress response, PIF, as crucial upstream factor, regulate the transcription of downstream target genes either independently or through interactions with other proteins. For example, the phyB-DET1-HFR1-PIF1 signaling pathway governs seed germination in response to light in *Arabidopsis* [63]; PIF4/TCP4–KRP1 restricts cell division and leaf size under high temperature conditions [64]; PIF4 transcriptionally represses *SPCH* influencing stomatal development [65]. Moreover, the FLS2–RBOHD–PIF4 module in *Arabidopsis* regulates responses to drought and salt stress [67]. In rice, OsPIL16 positively regulates *OsDREB1* expression, enhancing cold tolerance [39], while OsPIL13 and OsPIF14 induce growth by regulating cell wall organization and cell elongation genes in response to drought and salt stress [38, 66, 68]. These collective findings underscore the pivotal role of PIFs as integrators in biotic and abiotic stress signaling pathways, orchestrating plant growth and adaptation.

## 4. Materials and methods

### 4.1. Plant materials and abiotic stress treatments

We utilized seeds from the *Triticum aestivum* cultivar 'Chinese Spring' for all experiments in this study. To ensure cleanliness, the seeds underwent surface sterilization with 0.1% sodium hypochlorite, followed by a thorough rinse with deionized water. After sterilization, the seeds were strategically positioned on wet filter paper and allowed to germinate at a temperature of 24˚C for a duration of 4 days. Subsequently, the germinated seeds were transferred to a light incubator and cultivated under controlled conditions. The growth environment was maintained at a temperature of 15˚C, and a photoperiod of 16 hours light/8 hours dark was implemented. The nutrient supply for optimal growth was provided through Hoagland's nutrient solution, administered following established protocols [69].

Seedlings at the three-leaf stage underwent simulated drought treatments by immersing their roots in Hoagland's nutrient solution enriched with 20% polyethylene glycol (PEG)-6000. The exposure duration was meticulously controlled. Leaf samples were systematically collected at specific time points, namely 0, 1, 3, 6, 12, 24, and 48 hours post-treatment. Similar to the simulated drought protocol, ABA treatments were conducted using seedlings at the three-leaf stage. However, in this case, the Hoagland's nutrient solution was supplemented with 100 µM abscisic acid (ABA) instead of PEG. The same rigorous collection procedure was followed, involving time-point samplings, washing of leaves, swift freezing in liquid nitrogen, and storage at -80˚C for subsequent RNA extraction.

### 4.2. Genome-wide *PIF* identification in wheat and gene annotation

*PIF* genes were identified in the wheat genome using the IWGSC RefSeq v1.1 reference genome assembly (*https://wheat-urgi.versailles.inra.fr/*). To initiate the process, all eight known

*Arabidopsis* PIF protein sequences were downloaded from TAIR database (*http://www. arabidopsis.org/*) and used as queries in BLASTP searches of the wheat genome. The outcomes underwent rigorous curation to eliminate redundant sequences, then the presence of the conserved bHLH domain and APB motif were verified in the remaining sequences using SMART (*http://smart.embl.de/*). The subcellular localization of each protein was predicted using Cell-PLoc 2.0 (*http://www.csbio.sjtu.edu.cn/bioinf/plant-multi/*) [70]. The resulting TaPIF proteins underwent a thorough examination of their biochemical properties. These encompassed the grand average of hydropathicity (GRAVY), molecular weight (MW), theoretical isoelectric point (pI), and instability index. These properties were predicted with ExPasy (*http://web. expasy.org*) [71].

### 4.3. TaPIF phylogenetic and synteny analyses

The sequences of known PIF proteins in *Arabidopsis*, maize, rice, and *Brachypodium distachyon* were obtained from JGI (*https://phytozome.jgi.doe.gov/pz/portal.html*). Combining these sequences with the identified TaPIFs resulted in a dataset of 44 sequences. ClustalW [72] with default settings was employed to align the full-length amino acid sequences. The phylogenetic tree was constructed from using the Neighbor-Joining (NJ) method with 1000 bootstrap replicates in MEGA7 [73]. The resulting phylogenetic tree was visualized with Evolview (*https:// evolgenius.info//evolview-v2/*). Duplicate gene pairs among the PIFs were identified using the TBtools program MCScanx,employing an E-value threshold of $< 1 \times 10^{-10}$. Tandem duplicates were characterized when adjacent homologous TaPIF genes were present on a single chromosome with no more than one intervening gene. Segmental duplications, indicative of polyploidy or chromosome rearrangement, were also identified. Collinear relationships between *TaPIFs* and *PIF* genes in rice and *B. distachyon* were also analyzed with MCScanx using the default parameters. The visualization of chromosomal locations, duplications, and collinearity of *TaPIFs* were achieved using Circos [74] and TBtools [75].

### 4.4. Exon-intron structure and conserved motif analyses

The exon-intron structures of *TaPIFs* were determined based on the reference genome annotations. conserved protein motifs in TaPIFs were identified using the MEME suite (v5.3.3) with the following parameters: number of unique motifs set 5, motif length ranging from 6 to 50 amino acids, and the number of non-overlapping occurrences of each motif in a single sequence set to any. TBtoolswas used to visualize the gene structures [75].

### 4.5. *Cis*-acting element analysis

To determine potential functions of the *TaPIF* genes, we performed *cis*-regulatory element analysis of the promoter regions. Specifically, we focused on the 2-kb region upstream of the start codon was analyzed with the PlantCARE (*http://bioinformatics.psb.ugent.be/webtools/ plantcare/html/*) [76]. A heatmap constructed with TBtools [75] was used to visualize the distribution of key types of *cis*-acting elements among the *TaPIF* genes.

### 4.6. RNA-sequencing (RNA-seq) data analyse

To analyze spatiotemporal expression profiles of *TaPIF* genes, publicly-available expression data were obtained from WheatOmics 1.0 (*http://202.194.139.32/*) and WheatExp (*https:// wheat.pw.usda.gov/WheatExp/*) [77, 78]. For investigating TaPIF responses to biotic stress,. RNA-seq datasets for wheat infected with stripe rust (*Puccinia striiformis f. sp. Tritici*) or powdery mildew (*Blumeria graminis f. sp. Tritici*) were analyzed [79]. Similarly, *TaPIF* responses to

abiotic stress conditions and hormone treatments were analyzed using RNA-seq datasets for wheat treated with drought, heat, cold, salt, 6-benzylaminopurine (6-BA), ABA, GA, JA, and salicylic acid (SA) [80, 81]. Expression levels in all RNA-seq datasets were quantified in fragments per kilobase of transcript per million mapped reads (FPKM). Heatmaps illustrating the expression patterns were generated using TBtools.

## 4.7. RNA extraction and qRT-PCR

Expression levels of *TaPIF* genes were analyzed in response to simulated drought stress and ABA treatment as described above (section 4.1). The HiPure Plant RNA Kits (Guangzhou Magen Biotechnology Co., Ltd., R4151) was used to isolate total RNA from leaf tissue according to the manufacturer's instructions. Subsequently, cDNA was synthesized with the HiScript III 1st Strand cDNA Synthesis Kit (+gDNA wiper) (Vazyme Biotechnology Co., Ltd, R312). The qRT-PCR reaction system was carried out using the ChamQ SYBR Color qPCR Master Mix (Vazyme Biotechnology Co., Ltd, Q411) in the CFX 96 real-time PCR detection system (Bio-Rad, USA). Mean expression values were calculated from three biological replicates. *TaPIF* expression levels were normalized to the internal control gene *TaActin1* with the $2^{-\Delta\Delta Ct}$ method [82]. All primers are shown in S7 Table.

## 4.8. Subcellular localization analysis

To investigate the subcellular localization of TaPIF proteins, TaPIF–green fluorescent protein (GFP) fusion constructs were generated. The coding regions of *TaPIF2-1B* and *TaPIF4-5B* were amplified and individually fused to the region encoding the N-terminal end of GFP in the CaMV35S-GFP-NOS vector. Leaf tissue was collected from 2-week-old wheat seedlings and protoplasts were isolated as previously described, and the resulting protoplasts were transformed with a TaPIF-GFP plasmid or the empty vector using a PEG-mediated transfection protocol [69]. Fluorescent signals were visualized with an FV3000 laser-scanning confocal microscope (OLYMPUS, Japan).

## 4.9. Transcriptional sactivation assay

To assess the transactivation activity of TaPIF2-1B and TaPIF4-5B, the full coding sequences of *TaPIF2-1B* and *TaPIF4-5B* were amplified via PCR and each was inserted into the pGBKT7 vector. Chemically competent AH109 yeast cells (Shanghai Weidi Biotechnology Co., Ltd, YC1010) were transformed with the *TaPIF* construct or empty vector, plated on SD/-Trp media, and incubated for 3 d at 30°C. Subsequently, transformants were streaked onto several types of dropout media: SD/-Trp, SD/-Trp/-His/-Ade, and SD/-Trp/-His/-Ade/x-α-gal. The transactivation activity of each protein was evaluated based on yeast cell growth.

## 5. Conclusions

Global wheat production faces severe threats from climate change, particularly due to factors such as drought that significantly constrain food production. Understanding the roles and mechanisms of genes involved in stress response is essential for genetic improvement strategies aimed at achieving high yield and stress resistance in wheat. In this study, we identified 18 wheat *PIF* genes and investigated their collinearity, phylogenetic relationship, gene structure, protein motif compositions, expression patterns, and subcellular localization. The comprehensive analysis of *PIF* family members their involvement in wheat growth and development as well as stress response The findings of this study provide valuable insights that can guide further research into the functions of *TaPIFs*.

## Supporting information

**S1 Fig. Multiple sequence alignment of PIF proteins in *Arabidopsis*, rice, maize, *Brachypodium distachyon*, and wheat.**
(TIF)

**S2 Fig. Conserved motifs in TaPIF proteins.**
(TIF)

**S1 Table. Basic information about *PIF*s in wheat, *Arabidopsis*, rice, maize, and *Brachypodium distachyon*.**
(XLSX)

**S2 Table. Physicochemical properties of PIF proteins in *Arabidopsis*, rice, maize, and *Brachypodium distachyon*.**
(XLSX)

**S3 Table. The *Ka*/*Ks* ratios and duplication times of paralogous *TaPIF*s in wheat, rice, maize, and *Brachypodium distachyon*.**
(XLSX)

**S4 Table. Conserved motifs in TaPIF proteins.**
(XLSX)

**S5 Table. *Cis*-regulatory elements in *TaPIF* promoters.**
(XLSX)

**S6 Table. *TaPIF* expression levels in fragments per kilobase of transcript per million mapped reads (FPKM).**
(XLSX)

**S7 Table. Primers used in this study.**
(XLSX)

## Author Contributions

**Data curation:** Jian Wang.

**Software:** Zhen Guo.

**Supervision:** Tianqing Chen.

**Writing – original draft:** Hua Zhuang.

**Writing – review & editing:** Tianqing Chen.

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
