## [Decision Letter · Decision Letter 0]

29 Sep 2023

PONE-D-23-27525Genome-wide identification and comprehensive analysis of the phytochrome-interacting factor ( PIF ) gene family in wheatPLOS ONE

Dear Dr. Chen,

Thank you for submitting your manuscript to PLOS ONE. After careful consideration, we feel that it has merit but does not fully meet PLOS ONE’s publication criteria as it currently stands. Therefore, we invite you to submit a revised version of the manuscript that addresses the points raised during the review process.

Dear Dr Qing Tian Chen,

In this peer-review process, we invited two experts in this field, and both suggested that your manuscript be reconsidered for publication in PLOS ONE after addressing all the comments from the reviewers.

In my view, it should be careful to use the name "PIFs" for these genes since the present data are not enough to support this. It is better to mention these are "PIF candidates" in the manuscript. Otherwise, you need to support the data indicating the interactions between these proteins and Phytochrome in the examined species (wheat).

Thank you very much for submitting the work to PLOS ONE and we look forward to receiving your revised manuscript.

We look forward to receiving your revised manuscript.

Kind regards,

Nguyen Hoai Nguyen

Academic Editor

PLOS ONE

Journal Requirements:

https://www.sciencedirect.com/science/article/abs/pii/S0098847222004142?via%3Dihub

In your revision ensure you cite all your sources (including your own works), and quote or rephrase any duplicated text outside the methods section. Further consideration is dependent on these concerns being addressed.

3. Thank you for submitting the above manuscript to PLOS ONE. During our internal evaluation of the manuscript, we found significant text overlap between your submission and previous work in the [introduction, conclusion, etc.].

Please revise the manuscript to rephrase the duplicated text, cite your sources, and provide details as to how the current manuscript advances on previous work. Please note that further consideration is dependent on the submission of a manuscript that addresses these concerns about the overlap in text with published work.

[If the overlap is with the authors’ own works: Moreover, upon submission, authors must confirm that the manuscript, or any related manuscript, is not currently under consideration or accepted elsewhere. If related work has been submitted to PLOS ONE or elsewhere, authors must include a copy with the submitted article. Reviewers will be asked to comment on the overlap between related submissions (http://journals.plos.org/plosone/s/submission-guidelines#loc-related-manuscripts).]

We will carefully review your manuscript upon resubmission and further consideration of the manuscript is dependent on the text overlap being addressed in full. Please ensure that your revision is thorough as failure to address the concerns to our satisfaction may result in your submission not being considered further.

"This work was supported by the Shaanxi Province "High-level Talents Special Support Plan" Outstanding talents project, Construction of a shared platform for soil quality detection and evaluation in Shaanxi Province (2021PT-053)."

"This work was supported by the Shaanxi Province "High-level Talents Special Support Plan" Outstanding talents project, Construction of a shared platform for soil quality detection and evaluation in Shaanxi Province (2021PT-053)."

"This work was supported by the Shaanxi Province "High-level Talents Special Support Plan" Outstanding talents project, Construction of a shared platform for soil quality detection and evaluation in Shaanxi Province (2021PT-053)."

7. PLOS requires an ORCID iD for the corresponding author in Editorial Manager on papers submitted after December 6th, 2016. Please ensure that you have an ORCID iD and that it is validated in Editorial Manager. To do this, go to ‘Update my Information’ (in the upper left-hand corner of the main menu), and click on the Fetch/Validate link next to the ORCID field. This will take you to the ORCID site and allow you to create a new iD or authenticate a pre-existing iD in Editorial Manager. Please see the following video for instructions on linking an ORCID iD to your Editorial Manager account: https://www.youtube.com/watch?v=_xcclfuvtxQ

Reviewers' comments:

Reviewer's Responses to Questions

**Comments to the Author**

1. Is the manuscript technically sound, and do the data support the conclusions?

Reviewer #1: Yes

Reviewer #2: Partly

2. Has the statistical analysis been performed appropriately and rigorously? 

Reviewer #1: Yes

Reviewer #2: No

3. Have the authors made all data underlying the findings in their manuscript fully available?

Reviewer #1: Yes

Reviewer #2: Yes

4. Is the manuscript presented in an intelligible fashion and written in standard English?

Reviewer #1: Yes

Reviewer #2: Yes

5. Review Comments to the Author

Reviewer #1: The manuscript of Zhuang et al. presents the identification about the roles and expression patterns of phytochrome interacting factor (PIF) gene family in wheat. The authors systematically analyzed the functions of TaPIFs by bioinformatics, expression patterns, subcellular localization and transcriptional activation.

Overall, the evolutionary and bioinformatics analyses are extensive, well presented, supported by high-quality and well-organized figures. Some revisions are however necessary to add.

The manuscript is also in need of English language and grammar revisions that I will not report here, due to a too long list of indications.

Reviewer #2: This study by Zhuang et al. examined the PIF gene family in wheat using a variety of sequence-based analyses. The authors presented multiple results including phylogenetic analysis, gene and protein motif analysis, promoter cis-regulatory element comparison, gene expression data, and even subcellular localization results to support their study. In general, the findings have the potential to be a useful foundation for further studies involving TaPIFs.

Despite the comprehensive analysis by the authors, there are several critical questions that were not addressed in this version of the manuscript. Importantly, the authors need to show that the 34 TaPIFs are true PIF homologs. Arabidopsis PIFs are well-studied and contain several key features including a conserved bHLH domain as well as APB/APA domains. Furthermore, sequence alignment against other PIF homologs were not shown in this version of the manuscript, making it difficult to assess the accuracy of the identification. Moreover, the authors need to clarify why their phylogenetic analysis showed O. sativa, Z. mays, and Arabidopsis to have larger number of PIFs than what is currently known in literature.

Below are some major issues that need to be addressed:

(1) While the authors identified 34 TaPIFs via sequence similarity to known AtPIFs, the sequence alignment or similarity percentage/score were not provided in the manuscript. Hence, it is not possible to assess whether the identified proteins are truly homologs.

(2) The provision of instability indexes, GRAVY scores, and predicted subcellular localization is good information. To highlight the significance of these results, the authors can compare these indexes against all AtPIFs, which are well-studied.

(3) It is unclear how the authors conducted the phylogenetic analysis. In the introduction, the authors described O. sativa having only seven PIFs, Z. mays has eight PIFs, while Arabidopsis has nine PIFs. However, in Figure 1 and Table S1, these plant species were shown to have more PIFs than previously described. In Table S1, genes such as SPT, ALC, etc. were listed as a PIF gene. This is not true.

(4) The identification of common motifs amongst TaPIFs is compelling evidence to support the phylogeny analysis. To improve this data set, the authors should identify the presence of active phytochrome B-binding (APB) and active phytochrome A-binding (APA) motifs in the TaPIFs. APAs and APBs are defining features of PIFs, in addition to having conserved bHLH domains.

(5) The authors mentioned that motifs 1, 2, and 5 have the same arrangement in all TaPIFs, but did not further describe what these motifs contain and their significance. Is this the bHLH domain? If so, how do these domains compare with AtPIFs?

(6) Figures 6 and 7: It will be helpful to link the findings here to Figure 4 (cis-elements in promoter). This can elucidate if the cis-elements identified in Figure 4 result in the biotic and abiotic responses seen in Figures 6 and 7.

(7) Tests for statistical difference can be incorporated into the analyses, such as in the qRT-PCR results (Figure 8).

Below are some minor issues of this manuscript:

(1) In Figure 3, the intron-exon-UTR results should be presented first (either as panel A or rearranged to the left of the figure). This is because this dataset is referred to in the manuscript text before the motifs.

(2) Lines 197-198: TaPIF2 also has high expression in “leaves, flag leaves, and young spikes”, but it was omitted from this sentence.

(3) Lines 199-200: TaPIF6 was stated to be highly expressed in vegetative growth. However, its high expression is mostly limited to the roots; this alone should not constitute vegetative growth.

6. PLOS authors have the option to publish the peer review history of their article (what does this mean?). If published, this will include your full peer review and any attached files.

Reviewer #1: No

Reviewer #2: **Yes: **Benny Jian Rong Sng

---

## [Author Response · Author response to Decision Letter 0]

2 Nov 2023

Reply to the editors and reviewers

Reference: PONE-D-23-27525 

Title: Genome-wide identification and comprehensive analysis of the phytochrome-interacting factor (PIF) gene family in wheat

Authors: Hua Zhuang, Zhen Guo, Jian Wang, Tianqing Chen

Dear editors,

We thank you for your timely response to our submission and offer our thanks to the reviewers for their highly constructive, thoughtful comments and suggestions regarding our study. We have revised our article to avoid overlapping text and previous research, including introduction, conclusion, and other sections. We have revised the manuscript as appropriate based on their suggestions and have done our best to address all concerns raised by each reviewer. Enclosed, please find our point-by-point responses. We are very grateful for the editorial and scientific guidance provided in this review process, which has greatly improved the quality of our study and will inform our research going forward. We thank you kindly again for your consideration. Please let us know if any further information is needed. 

Response to Reviewer #1

The manuscript of Zhuang et al. presents the identification about the roles and expression patterns of phytochrome interacting factor (PIF) gene family in wheat. The authors systematically analyzed the functions of TaPIFs by bioinformatics, expression patterns, subcellular localization and transcriptional activation. Overall, the evolutionary and bioinformatics analyses are extensive, well presented, supported by highquality and well-organized figures. Some revisions are however necessary to add. The manuscript is also in need of English language and grammar revisions that I will not report here, due to a too long list of indications.

Response: Thank you for taking the time to review our manuscript. We appreciate your valuable feedback and suggestions, which have greatly contributed to improving the quality of our work. We have carefully considered each of your comments and made the necessary revisions accordingly. We have also sent the revised manuscript to a professional manuscript editing company staffed by American PhDs trained in the life sciences for correction and polishing of our English.

Response to Reviewer #2

This study by Zhuang et al. examined the PIF gene family in wheat using a variety of sequencebased analyses. The authors presented multiple results including phylogenetic analysis, gene and protein motif analysis, promoter cis-regulatory element comparison, gene expression data, and even subcellular localization results to support their study. In general, the findings have the potential to be a useful foundation for further studies involving TaPIFs.

Despite the comprehensive analysis by the authors, there are several critical questions that were not addressed in this version of the manuscript. Importantly, the authors need to show that the 34 TaPIFs are true PIF homologs. Arabidopsis PIFs are well-studied and contain several key features including a conserved bHLH domain as well as APB/APA domains. Furthermore, sequence alignment against other PIF homologs were not shown in this version of the manuscript, making it difficult to assess the accuracy of the identification. Moreover, the authors need to clarify why their phylogenetic analysis showed O. sativa, Z. mays, and Arabidopsis to have larger number of PIFs than what is currently known in literature.

Response: We thank the reviewer for their constructive comments and useful suggestions, the latter of which have helped us improve the manuscript. We re-identified TaPIF proteins in wheat according to the reviewer's suggestion, which were confirmed to contain conserved bHLH domain and APB domain one by one, and confirmed that a total of 18 TaPIF protein sequences belonged to 6 homeologous groups distributed on the A, B, and D subgenomes (see Page 5, Lines 86-88 and lines 89-91). Multiple sequences were compared for 44 PIF proteins from wheat, rice, Arabidopsis, Brachysteta dipaniculata, and maize to enhance the reliability of the results, and typical conserved domains were identified (Figure S1). In addition, we also corrected the PIF protein in Arabidopsis, rice, and maize, and reconstructed the phylogenetic tree (Figure 1A), and the corrected PIF protein number in each species was consistent with the references.

“A total of 18 TaPIF proteins with conserved domains of bHLH and APB were identified (Table 1, Figure S1).”

“The similarity percentage of TaPIF proteins were 96.4% (TaPIF1s), 97.1% (TaPIF2s), 94.8% (TaPIF3s), 90.4% (TaPIF4s), 91.7% (TaPIF5s), 93.1% (TaPIF6s), respectively.”

Below are some major issues that need to be addressed:

(1)While the authors identified 34 TaPIFs via sequence similarity to known AtPIFs, the sequence alignment or similarity percentage/score were not provided in the manuscript. Hence, it is not possible to assess whether the identified proteins are truly homologs.

Response: We appreciate the reviewer's suggestions. We have revised the section by providing sequence alignment in the manuscript, see Figure S1 .

(2)The provision of instability indexes, GRAVY scores, and predicted subcellular localization is good information. To highlight the significance of these results, the authors can compare these indexes against all AtPIFs, which are well-studied.

Response: We appreciate the reviewer's suggestions. The physicochemical properties of PIF protein in Arabidopsis, rice, Brachypodium distachyon and maize were added (see Table S1), and compare instability indexes, GRAVY scores, and predicted subcellular localization against TaPIFs for highlight the significance of these results, see Page 5, lines 99-103.

“In addition, we also predicted the above parameters of PIF protein in rice, Arabidopsis, Brachypodium distachyon, and maize (Table S1), finding these proteins were similar to TaPIF proteins on instability indexes, GRAVY scores, and subcellular localization. In particular, the hydrophilicity of PIF protein in Arabidopsis (-0.95 to 0.562) were generally higher than that in wheat.”

(3)It is unclear how the authors conducted the phylogenetic analysis. In the introduction, the authors described O. sativa having only seven PIFs, Z. mays has eight PIFs, while Arabidopsis has nine PIFs. However, in Figure 1 and Table S1, these plant species were shown to have more PIFs than previously described. In Table S1, genes such as SPT, ALC, etc. were listed as a PIF gene. This is not true.

Response: We appreciate the reviewer's suggestions. We mistakenly included members of all 15 subgroups of the bHLH superfamily when in fact there were members that did not belong to the PIF protein. We carefully conducted amino acid multiple sequence alignment and reconstructed the developmental tree, and excluded the members of the bHLH domain and APB domain that did not have the typical conserved PIF, and revised in Figure 1 and Table S1. At the same time, the reference in the preface is corrected to make the preceding. 

(4)The identification of common motifs amongst TaPIFs is compelling evidence to support the phylogeny analysis. To improve this data set, the authors should identify the presence of active phytochrome B-binding (APB) and active phytochrome A-binding (APA) motifs in the TaPIFs. APAs and APBs are defining features of PIFs, in addition to having conserved bHLH domains.

Response: We appreciate the reviewer's suggestions. We carefully performed multiple amino acid alignment and mapped the typically conserved bHLH and APB domains of PIF on the TaPIF amino acid sequence of wheat, See Figure S1.

(5)The authors mentioned that motifs 1, 2, and 5 have the same arrangement in all TaPIFs, but did not further describe what these motifs contain and their significance. Is this the bHLH domain? If so, how do these domains compare with AtPIFs?

Response: We appreciate the reviewer's suggestions. We describe what these motifs contain in the manuscript, see Page 7, lines 145-146.

“All of the TaPIFs contained the same arrangement of motif 2 and motif 5, which represent bHLH and APB domains, the features of PIF proteins, respectively.”

(6)Figures 6 and 7: It will be helpful to link the findings here to Figure 4 (cis-elements in promoter). This can elucidate if the cis-elements identified in Figure 4 result in the biotic and abiotic responses seen in Figures 6 and 7.

Response: We are grateful for the reviewer's suggestion. We analyzed and added the relationship between cis-elements identified in Figure 4 and the result in the biotic and abiotic responses seen in Figures 6 and 7 according to the reviewer's suggestions, see Page 10, lines 203-207; lines 212-215, and Page 10, lines 219-222 and Page 11, lines 226-235.

“Combined with the above cis-acting element identification, the promoter regions of TaPIF1-1A/B/D, TaPIF2-1A/B/D, TaPIF5-5A/B/D, and TaPIF6-5A/B/D identified defense and stress responses, SA response, or MeJA response elements (Figure 4, Table S5), which may be the reason for the response of these genes to the infection of stripe rust or powdery mildew.”

“Under drought conditions, TaPIF1-1A/B/D were up-regulated compared with control plants; there were no significant changes in TaPIF4 expression levels, and the remainder of TaPIF genes were down-regulated (Figure 7A, Table S6), which consistent with their promoter region containing drought-response elements like MBS or DRE (Figure 4).”

“Besides, TaPIF1-1A/B/D, TaPIF2-1A/B/D, and TaPIF6-5D were up-regulated under cold conditions, whereas TaPIF3-2D, TaPIF4-5B/D, and TaPIF5-5A/B/D were down-regulated (Figure 7B, Table S6), which may be due to the existence of LTR element in promoter region of TaPIF1-1B, TaPIF2-2A/D, TaPIF3-2A, TaPIF5-5B/D, TaPIF6-5D (Figure 4).”

“TaPIF3-2A/B, TaPIF4-5A/B/D, TaPIF5-5A/B/D, and TaPIF6-5A/B/D were down-regulated, while TaPIF2-1B/D were up-regulated in response to SA treatment, although only TaPIF3-2A, TaPIF5-5D, and TaPIF6-5D had SA-responsive elements in their promoters. The expression of all TaPIFs were inhibited by 6-BA treatment; TaPIF1-1A/B, TaPIF2-1A/B/D were down-regulated, while TaPIF3-2A/B/D, TaPIF4-5A/B/D, and TaPIF5-5A/B/D were up-regulated in response to ABA exposure compare to control, which were inseparable from the ABRE-element of their promoters (Figure 7C, Table S6). For GA and JA treatments, except that the expression level of TaPIF5-5A/B/D had no significance, the remaining TaPIF genes were down-regulated under GA treatments; TaPIF2-1A/B/D, TaPIF4-5A/B/D, and TaPIF5-5A/B/D had obvious response to exposure to JA (Figure 7C, Table S6).”

(7)Tests for statistical difference can be incorporated into the analyses, such as in the qRT-PCR results (Figure 8). 

Response: According to reviewer's suggestion, we conducted statistical difference in the qRT-PCR, and the analytical method and significance of the data are described in the figure notes. See Figure 8 and Page lines 711-712. 

“The asterisks indicate significant differences based on the results of the one-way ANOVA test (* P < 0.05; ** P < 0.01).”

Below are some minor issues of this manuscript:

(1)In Figure 3, the intron-exon-UTR results should be presented first (either as panel A or rearranged to the left of the figure). This is because this dataset is referred to in the manuscript text before the motifs.

Response: We appreciate the reviewer's suggestion. We have rearranged the intron- exon-UTR results to the left of the Figure 3. 

(2)Lines 197-198: TaPIF2 also has high expression in “leaves, flag leaves, and young spikes”, but it was omitted from this sentence.

Response: According to the reviewer's suggestion, the omitted description has been added to Page 9, Lines 188-191.

“TaPIF1s had high expression in stem, flag leaves, and young spikes; TaPIF2s has high expression in leaves, flag leaves, and young spikes; TaPIF3s, TaPIF4s, and TaPIF5s were highly expressed in the stem, leaves, flag leaves, and young spikes; TaPIF6s were most highly expressed in leaves and flag leaves.”

(3)Lines 199-200: TaPIF6 was stated to be highly expressed in vegetative growth. However, its high expression is mostly limited to the roots; this alone should not constitute vegetative growth.

Response: The authors would like to thank the reviewer for their time and careful attention to detail. We have revised the description to make the conclusion more accurate, see Page 9, Line 199.

“ TaPIF6s were most highly expressed in leaves and flag leaves.”

Overall, we have made several modifications to the manuscript and done our best to thoroughly address all of the reviewers’ concerns. We again thank the editor and reviewers for their invaluable critique, comments, and suggestions, all of which have helped to improve our manuscript.

We kindly thank you again for your consideration!

Sincerely yours, 

Dr. Tianqing Chen

Shaanxi Provincial Land Engineering Construction Group Co., Ltd. Xi'an, China.

---

## [Decision Letter · Decision Letter 1]

20 Nov 2023

PONE-D-23-27525R1Genome-wide identification and comprehensive analysis of the phytochrome-interacting factor ( PIF ) gene family in wheatPLOS ONE

Dear Dr. Chen,

Thank you for submitting your manuscript to PLOS ONE. After careful consideration, we feel that it has merit but does not fully meet PLOS ONE’s publication criteria as it currently stands. Therefore, we invite you to submit a revised version of the manuscript that addresses the points raised during the review process.

Based on both Reviewers' comments, I suggest that this work has to be revised again and returned for review afterward.

**Please consider both Reviewers' comments carefully and make sure that you revise all the comments carefully and thoroughly.**

Thank you very much for submitting your work to PLOS ONE.

We look forward to receiving your revised manuscript.

Kind regards,

Nguyen Hoai Nguyen

Academic Editor

PLOS ONE

Journal Requirements:

Additional Editor Comments:

Dear Dr. Chen,

Based on both Reviewers' comments, I suggest that this work has to be revised again and returned for review afterward.

Please consider both Reviewers' comments carefully and make sure that you revise all the comments carefully and thoroughly.

Thank you very much for submitting your work to PLOS ONE.

Best regards,

Nguyen

Reviewers' comments:

Reviewer's Responses to Questions

**Comments to the Author**

1. If the authors have adequately addressed your comments raised in a previous round of review and you feel that this manuscript is now acceptable for publication, you may indicate that here to bypass the “Comments to the Author” section, enter your conflict of interest statement in the “Confidential to Editor” section, and submit your "Accept" recommendation.

Reviewer #1: (No Response)

Reviewer #2: (No Response)

2. Is the manuscript technically sound, and do the data support the conclusions?

Reviewer #1: Partly

Reviewer #2: Yes

3. Has the statistical analysis been performed appropriately and rigorously? 

Reviewer #1: I Don't Know

Reviewer #2: Yes

4. Have the authors made all data underlying the findings in their manuscript fully available?

Reviewer #1: Yes

Reviewer #2: Yes

5. Is the manuscript presented in an intelligible fashion and written in standard English?

Reviewer #1: Yes

Reviewer #2: No

6. Review Comments to the Author

Reviewer #1: There is a fundamental and serious error in the manuscript, namely that the number of the latest defined Arabidopsis PIFs is 8, not 7 (Lee and Choi 2017, jiang et al., 2022). Authors need to refer to the latest literature results, not those from 10 years ago (Leivar and Quail, 2011). All subsequent results and conclusions are therefore out of the question, and this error was pointed out last time.

Reviewer #2: I greatly appreciate the authors’ efforts in revising the manuscript according to my previous suggestions. The authors have addressed most of my major concerns. The data and conclusions in this study are now generally convincing. However, due to some errors in the writing and figures, the manuscript is unfortunately not ready for publication.

Below is a list of minor concerns that the authors should address:

(1) Line 92: There is no “TaPIF7-5D” after the reanalysis.

(2) Line 102: The number “0.562” should be written as “-0.562”, otherwise “-0.95 to 0.562” will be interpreted as ranging from a negative to positive number.

(3) Line 108: Instead of “ZmPIFs and BdPIFs”, did the authors mean “OsPIFs and ZmPIFs”? In figure 1A, BdPIFs distributed across all four groups.

(4) Line 115: “Identified” should not itacilized.

(5) Line 121-122: Typo error, as “18 collinear pairs” was repeated.

(6) Line 122: “Genome-wide replication” is abbreviated wrongly.

(7) Line 127: Additional comma.

(8) Line 127-128: “PIF genes had been subject to strong purifying selection In rice and B. distachyon” should be substantiated with data or by citing a reference. Similarly, the claim that “similar number of gene duplication events were found between B. distachyon, O. sativa and wheat, respectively” should be supported with evidence on the number of duplication events for each comparison.

(9) Line 145: In Figure S2, motif 2 and 5 do not match with the known sequences of bHLH and APB domains, respectively. This part should be revised after careful analysis of the results. By comparing with published consensus sequences, motif 1 appears to be the bHLH domain while motif 2 matches closely with the APB domain. The authors can demonstrate this by comparing the identified motif sequences in figure S2 with published bHLH and APB domain sequences (via sequence alignment).

(10) The authors should note that for AtPIFs, the APB domain tends to be in the N-terminus while the bHLH domain is found in the C-terminus. This is not conserved for all Arabidopsis bHLHs. Since, this pattern (APB domain in N-terminus and bHLH domain in C-terminus) is also observed in all TaPIFs, it strongly supports the claim that the identified proteins are functionally similar to Arabidopsis PIFs. The authors can consider referring to “The Basic Helix–Loop–Helix Transcription Factor Family in Plants: A Genome-Wide Study of Protein Structure and Functional Diversity” by Heim et al. (2003) (Molecular Biology and Evolution).

(11) Line 251-253: “Transactivation assays were performed in yeast with TaPIF2 and TaPIF4 .Each gene was cloned from the wheat cultivar ‘Chinese Spring’”. Which TaPIF2 gene was cloned – TaPIF2-1A, TaPIF2-1B, or TaPIF2-1D? Likewise, which TaPIF4 gene was cloned?

(12) Line 266-268: Please revise this sentence. While the meaning is generally understood, the sentence is too long and has too many commas, making it unclear.

(13) Line 269: “whole genome replication” was abbreviated wrongly.

(14) Lines 272-273: “Here, collinearity analysis of wheat, O. sativa, and B. distachyon revealed 21, seven, 24, and 25 collinearity PIF gene pairs, respectively”. There are three plant species and four collinearity PIF gene pairs. Please clarify this.

(15) Lines 334-336: The evidence shown in Figure 6B did not include all three genes for TaPIF2 nor TaPIF4. It is therefore erroneous to claim that the subcellular localization assay confirmed the location of all three TaPIF2 and TaPIF4.

(16) Figure legends for Figure 2B: Please state what the orange and green bars represent.

(17) Figure 7: TaPIFs included TaPIF7 and TaPIF8, which were not part of the list of TaPIFs after the re-analysis. The manuscript text does not correspond to the figure.

(18) Figure legends for Figure 7: Please state what the color bars represent. Are they z-scores or relative expression?

(19) Figure 8: The time points for TaPIF2 (both PEG and ABA treatment) are incorrect. The time points are “3, 6, 12, 24, 12, 24h”.

(20) Table S7: Last primer is labelled as TaPIF5_GFP. Is this a typo error and that it should be TaPIF4?

(21) Generally, the writing can be further revised to improve sentence structure. Some sentences were too long with multiple punctuation, which makes it difficult to read. There are also multiple labelling and typo errors throughout the manuscript (there are additional typo errors that are not specifically mentioned in this review). The authors should thoroughly check the writing before resubmission.

7. PLOS authors have the option to publish the peer review history of their article (what does this mean?). If published, this will include your full peer review and any attached files.

Reviewer #1: No

Reviewer #2: **Yes: **Benny Jian Rong Sng

---

## [Author Response · Author response to Decision Letter 1]

5 Dec 2023

Dear editors,

Thank you for your thorough review and patient guidance and offer our thanks to the reviewers for their thoughtful comments and suggestions regarding our study. We have carefully read and addressed reviewers’ suggestions in the manuscript and have done our best to address all concerns raised. Special attention has been given to rectifying details and made every effort to minimize errors in spelling and grammar, striving to enhance the overall quality of the paper. We are very grateful for the editorial and scientific guidance provided in this review process, which has greatly improved the quality of our study and will inform our research going forward. Once again, we appreciate your valuable feedback, and we look forward to any further guidance you may provide.

Response to Reviewer #1

There is a fundamental and serious error in the manuscript, namely that the number of the latest defined Arabidopsis PIFs is 8, not 7 (Lee and Choi 2017, jiang et al., 2022). Authors need to refer to the latest literature results, not those from 10 years ago (Leivar and Quail, 2011). All subsequent results and conclusions are therefore out of the question, and this error was pointed out last time.

Response: The authors appreciate the reviewer's suggestions. We understand the concerns about the accuracy of the reference sequence, fearing that the identification of PIF family members in wheat may be incomplete. In response to the reviewer's suggestions, we referenced the work of Lee and Choi (2017) and included the previously overlooked Arabidopsis AtPIF2 (AT2G46970) protein. We employed BLASTp to search for proteins in wheat with high similarity to AtPIF2, using a criterion of alignment length greater than 80 amino acids and similarity exceeding 50%. The identified proteins are TraesCS5A02G376500, TraesCS5B02G380200, TraesCS5D02G386500, TraesCS1A02G083000, TraesCS1B02G100400, and TraesCS1D02G084200, corresponding to our designated TaPIF5-5A, TaPIF5-5B, TaPIF5-5D, TaPIF1-1A, TaPIF1-1B, and TaPIF1-1D. These results did not alter the number of TaPIF family members identified in our previous characterization of wheat. Based on these findings, we incorporated AtPIF2 for multiple sequence alignment and phylogenetic analysis, subsequently revising Figure 1 and Figure S1.

“These include eight identified PIFs in tomato and apple [11, 12], respectively, and six identified PIFs in rice (Oryza sativa) and Brachypodium distachyon [13, 14], eight reported PIFs in Arabidopsis [11], potato [15], and maize (Zea mays) [16], respectively ”

Response to Reviewer #2

I greatly appreciate the authors’ efforts in revising the manuscript according to my previous suggestions. The authors have addressed most of my major concerns. The data and conclusions in this study are now generally convincing. However, due to some errors in the writing and figures, the manuscript is unfortunately not ready for publication.

Response: Thank you for your thoughtful review and constructive feedback. We appreciate your acknowledgment of our efforts in revising the manuscript and addressing your major concerns and glad to hear that the data and conclusions are generally convincing. We apologize for any remaining errors in writing and figures that may have hindered the publication readiness of the manuscript. We will carefully review and rectify these issues promptly to ensure the overall quality of the paper. Your insights are invaluable, and we are committed to delivering a polished and error-free manuscript. Once again, thank you for your time and feedback. We look forward to submitting the revised version for your consideration.

Below is a list of minor concerns that the authors should address:

(1)Line 92: There is no “TaPIF7-5D” after the reanalysis.

Response: The authors are very grateful for the reviewer's meticulous review, which helped identify our oversights. We have revised the corresponding descriptions to align them with the results of the reanalysis (Page 5, Line 90).

“TaPIF protein length ranged from 336 (TaPIF4-5A) to 516 (TaPIF5-5D) amino acids in length .”

(2)Line 102: The number “0.562” should be written as “-0.562”, otherwise “-0.95 to 0.562”will be interpreted as ranging from a negative to positive number.

Response: We appreciate the reviewer's suggestions. We have made the necessary corrections in the revised manuscript to avoid any misunderstandings (Page 5, Line 102).

“the hydrophilicity of PIF protein in Arabidopsis (-0.95 to -0.562) were generally higher than that in wheat.”

(3)Line 108: Instead of “ZmPIFs and BdPIFs”, did the authors mean “OsPIFs and ZmPIFs”? In figure 1A, BdPIFs distributed across all four groups.

Response: We appreciate the reviewer's suggestions, and we have revised the corresponding section accordingly (Page 6, Line 107). 

“TaPIFs and BdPIFs were distributed among all four groups, ZmPIFs and OsPIFs were clusterd in group I, group III, and group IV.”

(4)Line 115: “Identified” should not itacilized.

Response: The authors appreciate the reviewer's attention to important details, and they have modified them to the correct format in the revised manuscript (Page 6, Line 114).

“TaPIF genes that were identified could be clustered into 6 homoeologous groups in wheat.”

(5)Line 121-122: Typo error, as “18 collinear pairs” was repeated.

Response: We appreciate the reviewer's suggestions, and repeated characters have been removed in the revised manuscript (Page 6, Line 121-122).

“These duplication events resulted in 23 collinear pairs of TaPIFs in the wheat genome, of which 18 collinear pairs were due to genome-wide duplication (WGD) (Figure 2A; Table S3) .”

(6)Line 122: “Genome-wide replication” is abbreviated wrongly.

Response: The authors appreciate the reviewer's suggestions, and corrections have been made in the revised manuscript to ensure consistency between the full term and its abbreviation (Page 6, Line 119). 

“These gene replication events resulted in 23 collinear pairs of TaPIFs in the wheat genome, of which 18 collinear pairs were due to genome-wide duplication (WGD).”

(7)Line 127: Additional comma.

Response: We appreciate the reviewer's suggestions. We have already made the corrections.

“Notably, chromosomes 5A/B/D of wheat exhibited the highest homology with O. sativa and B. distachyon. PIF genes had been subject to strong purifying selection in rice and B. distachyon as their Ka/Ks ratios are less than 1 [43]”

(8)Line 127-128:“PIF genes had been subject to strong purifying selection in rice and B. distachyon”should be substantiated with data or by citing a reference. Similarly, the claim that“similar number of gene duplication events were found between B. distachyon, O. sativa and wheat, respectively”should be supported with evidence on the number of duplication events for each comparison.

Response: We appreciate the reviewer's suggestions. We have made modifications and additions to enhance the credibility of our descriptions.

“Notably, chromosomes 5A/B/D of wheat exhibited the highest homology with O. sativa and B. distachyon. PIF genes had been subject to strong purifying selection in rice and B. distachyon as their Ka/Ks ratios are less than 1 [43], and the similar number of gene duplication events was found in wheat/O. sativa (24) and wheat/B. distachyon (25), respectively.”

(9)Line 145: In Figure S2, motif 2 and 5 do not match with the known sequences of bHLH and APB domains, respectively. This part should be revised after careful analysis of the results. By comparing with published consensus sequences, motif 1 appears to be the bHLH domain while motif 2 matches closely with the APB domain. The authors can demonstrate this by comparing the identified motif sequences in figure S2 with published bHLH and APB domain sequences (via sequence alignment).

Response: We appreciate the reviewer's suggestions. After careful comparison and verification, we have made modifications in the revised manuscript to ensure the accuracy and reliability of our description (Page 7, Lines 146-147).

“All of the TaPIFs contained the same arrangement of motif 1 and motif 2, representing bHLH and APB domains, the features of PIF proteins, respectively.”

(10)The authors should note that for AtPIFs, the APB domain tends to be in the N-terminus while the bHLH domain is found in the C-terminus. This is not conserved for all Arabidopsis bHLHs. Since, this pattern (APB domain in N-terminus and bHLH domain in C-terminus) is also observed in all TaPIFs, it strongly supports the claim that the identified proteins are functionally similar to Arabidopsis PIFs. The authors can consider referring to “The Basic Helix–Loop–Helix Transcription Factor Family in Plants: A Genome-Wide Study of Protein Structure and Functional Diversity” by Heim et al. (2003) (Molecular Biology and Evolution).

Response: The authors are deeply grateful for the valuable suggestions provided by the reviewer, which have been instrumental in enhancing the quality of our work. We have added further description in this section in the discussion to make our analysis more comprehensive (Page 13, Lines 284-287).

“Additionally, we have observed that the APB domain and bHLH domain tend to be located at the N-terminus and C-terminus in AtPIF proteins, respectively (Figure S2). This similar arrangement pattern is also observed in all TaPIFs proteins, which strongly supports the claim that the identified TaPIF proteins were functionally similar to AtPIF proteins [49].”

(11)Line 251-253:“Transactivation assays were performed in yeast with TaPIF2 and TaPIF4 .Each gene was cloned from the wheat cultivar ‘Chinese Spring’”. Which TaPIF2 gene was cloned – TaPIF2-1A, TaPIF2-1B, or TaPIF2-1D? Likewise, which TaPIF4 gene was cloned?

Response: The authors appreciate the reviewer's suggestions. We cloned TaPIF2-1D and TaPIF4-5A as a representative to verify their transcriptional activation and subcellular localization due to the high similarity among homologous genes as mentioned earlier (see Page 5, lines 88-90). In response to the reviewer's comments, we have supplemented this information in the revised manuscript (see Page 12, lines 250-254; Lines 258-260) and made corresponding modifications in Figure 9.

 “Due to the high similarity among homologous genes, One copy of homologous was selected for experimental validation. TaPIF2-1B and TaPIF4-5B were cloned from the wheat cultivar ‘Chinese Spring’, inserted into the vector pGBKT7, and transformed into yeast. TaPIF2-1B and TaPIF4-5B promoted increased yeast growth on SD/-Trp/-His/-Ade medium compared to yeast transformed with the empty vector negative control .”

“we developed transient expression vectors to express plasmids encoding TaPIF-GFP fusion proteins (TaPIF2-1B-GFP and TaPIF4-5B-GFP) in wheat protoplasts.”

(12)Line 266-268: Please revise this sentence. While the meaning is generally understood, the sentence is too long and has too many commas, making it unclear.

Response: We appreciate the reviewer's suggestions. We have revised the sentence to make it clear and concise (Page 12, Lines 266-269).

“In this study, we identified 18 PIF genes in wheat, categorized into 6 homeologous groups (Table 1), which is a greater number of PIF genes than in the Arabidopsis, rice, maize, or B. distachyon genomes (Table S1). This increase is likely due to wheat being a heterohexaploid with three subgenomes.”

(13)Line 269: “whole genome replication” was abbreviated wrongly.

Response: The authors appreciate the reviewer's suggestions, and corrections have been made in the revised manuscript (Page 12, Lines 269-270)

 “Polyploid events or whole genome duplication (WGD) and amplification events play an important role in the genome evolution of plant species .”

(14)Lines 272-273: “Here, collinearity analysis of wheat, O. sativa, and B. distachyon revealed 21, seven, 24, and 25 collinearity PIF gene pairs, respectively”. There are three plant species and four collinearity PIF gene pairs. Please clarify this.

Response: We appreciate the reviewer's suggestions. We have corrected this error in the revised manuscript (Page 13, Lines 272-274).

“Here, collinearity analysis of wheat/wheat, wheat/O. sativa, and wheat/B. distachyon revealed 21, 24, and 25 collinearity PIF gene pairs, respectively.”

(15)Lines 334-336: The evidence shown in Figure 6B did not include all three genes for TaPIF2 nor TaPIF4. It is therefore erroneous to claim that the subcellular localization assay confirmed the location of all three TaPIF2 and TaPIF4.

Response: We appreciate the reviewer's suggestions. As suggested by the reviewer in point 11 (see Page 5, lines 89-91; Page 12, lines 255-258, Line 264 ), we have made the necessary corrections in the main text (see Page 16, lines 345-368) and Figure 9 .

“In vitro transcriptional activity assays with TaPIF2-1D and TaPIF4-5A, provided empirical evidence that at least some TaPIFs serve as transcriptional activators (Figure 9A). Furthermore, subcellular localization assays confirmed that the three TaPIF2-1D and TaPIF4-5A were localized to the nucleus as predicted (Figure 9B) ”.

(16)Figure legends for Figure 2B: Please state what the orange and green bars represent.

Response: The authors are grateful for the reviewer's suggestions. We have incorporated the corresponding descriptions in the figure legends for better reader comprehension (Page 30, lines 679-681).

“The orange bars represent chromosomes in wheat, while the green bars represent chromosomes in rice and B. distachyon. The text on the sides of the bars indicates the chromosome numbers.”

(17)Figure 7: TaPIFs included TaPIF7 and TaPIF8, which were not part of the list of TaPIFs after the re-analysis. The manuscript text does not correspond to the figure.

Response: We appreciate the reviewer's suggestions. The authors apologize for the oversight, and the accurate names have been rectified in Figure 8.

(18)Figure legends for Figure 7: Please state what the color bars represent. Are they z-scores or relative expression?

Response: We appreciate the reviewer's suggestions. We have added explanatory notes in the figure legend of Figure 6 and 7 to clarify the meaning represented by the color bars (Page 28, lines 708-709; Page 30, lines 719-721).

“The color bars represent the level of gene expression, with blue indicating low expression and red indicating high expression.”

(19)Figure 8: The time points for TaPIF2 (both PEG and ABA treatment) are incorrect. The time points are “3, 6, 12, 24, 12, 24h”.

Response: The authors are sincerely grateful to the reviewer for identifying this crucial error. We have made the necessary corrections in Figure 8.

(20)Table S7: Last primer is labelled as TaPIF5_GFP. Is this a typo error and that it should be TaPIF4?

Response: We appreciate the reviewer for pointing out this error. We have rectified it in Table S7.

(21)Generally, the writing can be further revised to improve sentence structure. Some sentences were too long with multiple punctuation, which makes it difficult to read. There are also multiple labelling and typo errors throughout the manuscript (there are additional typo errors that are not specifically mentioned in this review). The authors should thoroughly check the writing before resubmission.

Response: Thank you for your thoughtful review and constructive feedback on our manuscript. We appreciate your insights into the need for further improvement in sentence structure. We acknowledge your observations regarding lengthy sentences with multiple punctuations and the presence of labeling and typo errors throughout the manuscript. We have take your comments to heart and conduct a thorough review of the writing to address the mentioned issues and additionally submitted the revised manuscript to a professional editing service to correct and polish the English. Your feedback is invaluable in enhancing the overall quality of our paper.

Overall, we have made several modifications to the manuscript and done our best to thoroughly address all of the reviewers’ concerns. We again thank the editor and reviewers for their invaluable critique, comments, and suggestions, all of which have helped to improve our manuscript.

We kindly thank you again for your consideration!

Sincerely yours, 

Dr. Tianqing Chen

Shaanxi Provincial Land Engineering Construction Group Co., Ltd. Xi'an, China.

---

## [Editor Report · Decision Letter 2]

10 Dec 2023

Genome-wide identification and comprehensive analysis of the phytochrome-interacting factor ( PIF ) gene family in wheat

PONE-D-23-27525R2

Dear Dr. Chen,

We’re pleased to inform you that your manuscript has been judged scientifically suitable for publication and will be formally accepted for publication once it meets all outstanding technical requirements.

Kind regards,

Nguyen Hoai Nguyen

Academic Editor

PLOS ONE
---

## [Editor Report · Acceptance letter]

27 Dec 2023

PONE-D-23-27525R2 

PLOS ONE

Dear Dr. Chen, 

I'm pleased to inform you that your manuscript has been deemed suitable for publication in PLOS ONE. Congratulations! Your manuscript is now being handed over to our production team.

Kind regards, 

on behalf of

Dr. Nguyen Hoai Nguyen 

Academic Editor

PLOS ONE